# Cholangiocytes contribute to hepatocyte regeneration after partial liver injury during growth spurt in zebrafish

Sema Elif Eski [1], Jiarui Mi [2,3], Macarena Pozo-Morales[1,12], Gabriel Garnik Hovhannisyan [4], Camille Perazzolo[1], Rita Manco [5], Imane Ez-Zammoury[6], Dev Barbhaya[7], Anne Lefort[1], Frédérick Libert[1], Federico Marini [8,9], Esteban N. Gurzov [4], Olov Andersson [2,10] & Sumeet Pal Singh [1,11] ✉

The liver's regenerative ability depends on injury extent. Minor injuries are repaired by hepatocyte self-duplication, while severe damage triggers cholangiocyte involvement in hepatocyte recovery. This paradigm is well-documented for adult animals but is less explored during rapid growth. We design two partial liver injury models in zebrafish, which were investigated during growth spurts: 1) partial ablation, killing half the hepatocytes; and 2) partial hepatectomy, removing half a liver lobe. In both injuries, de novo hepatocytes emerged alongside existing ones. Single-cell transcriptomics and lineage tracing with Cre-driver lines generated by genome editing identified cholangiocytes as the source of de novo hepatocytes. We further identify active mTORC1 signalling in the uninjured liver of growing animal to be a regulator of the enhanced plasticity of cholangiocytes. Our study suggests cholangiocyte-to-hepatocyte transdifferentiation as the primary mechanism of liver regeneration during periods of rapid growth.

Liver regeneration after minor injuries, including surgical resection of even up to two-thirds of the liver mass, is driven by hepatocyte proliferation as the primary mechanism[1–7]. However, if hepatocyte proliferation is impaired or is insufficient (due to extensive loss of hepatocytes), cholangiocytes (also known as biliary epithelial cells (BECs) or ductal cells) can transdifferentiate into hepatocytes to aid in regeneration[8–17]. Notably, the conversion of cholangiocytes to hepatocytes has been documented in the liver of human patients suffering from chronic liver diseases[18–21]. This backup repair mechanism, known as 'facultative regeneration'[9,10], could be crucial in clinical settings for recovery from acute liver failure, such as in fulminant hepatitis and cirrhosis.

As per the current paradigm of liver regeneration, the capacity of hepatocytes to switch from quiescence to cell cycle dictates the

[1]Laboratory of Regeneration and Stress Biology, Institut de Recherche Interdisciplinaire en Biologie Humaine et Moléculaire (IRIBHM-Jacques E. Dumont), Université libre de Bruxelles, Brussels, Belgium. [2]Department of Cell and Molecular Biology, Karolinska Institutet, 17177 Stockholm, Sweden. [3]Department of Gastroenterology, Sir Run Run Shaw Hospital, School of Medicine, Zhejiang University, Hangzhou, Zhejiang, China. [4]Signal Transduction and Metabolism Laboratory, Université libre de Bruxelles, Anderlecht, 1070 Brussels-Capital Region, Belgium. [5]Laboratory of Hepato-gastroenterology, Institut de Recherche Expérimentale et Clinique, Université Catholique de Louvain, Brussels, Belgium. [6]Faculty of Pharmacy and Biomedical Sciences, Université Catholique de Louvain, Brussels, Belgium. [7]Indian Institute of Technology Kanpur (IIT-Kanpur), Kanpur, India. [8]Institute of Medical Biostatistics, Epidemiology and Informatics (IMBEI), University Medical Center of the Johannes Gutenberg University Mainz, Mainz, Germany. [9]Research Center for Immunotherapy (FZI), Mainz, Germany. [10]Department of Medical Cell Biology, Uppsala University, Biomedical Centre, Uppsala, Sweden. [11]Department of Life Sciences, School of Natural Sciences, Shiv Nadar Institution of Eminence, Delhi, India. [12]Present address: Molecular Oncology, Spanish National Cancer Research Centre (CNIO), Madrid, Spain. ✉e-mail: sumeet.pal.singh@ulb.be

cellular nature of regeneration in the liver. In an adult liver, hepatocytes are predominantly quiescent[22,23], with mitosis being observed in 1 out of every 20,000 hepatocytes in mouse[24]. In response to minor injuries in adult, a subset of quiescent hepatocytes can re-enter the cell cycle to replace the lost hepatocytes. But, in a growing animal, hepatocytes have a high proliferative baseline as cell-cycle is required to increase the size of the organ. For instance, in mouse, large numbers of cycling hepatocytes are observed from postnatal day 5 to 25, with 30% of hepatocytes cycling on postnatal day 10[25]. In the contexts of high baseline proliferation, it is unclear if hepatocytes can enhance their cellular output in response to minor injuries. Understanding the cellular mode of liver regeneration in growing animals can help tailor regenerative therapies for pediatric and juvenile liver disorders.

In this study, we evaluate the intersection between growth and regeneration. Particularly, we study the cellular response of the liver to partial injuries during a growth spurt in zebrafish. To induce partial injuries, we developed two protocols: 1) a genetic mosaic system for inducible ablation of a sub-set of hepatocytes; and 2) surgical resection of the liver. Using a combination of in vivo live imaging, single-cell transcriptomics and genetic lineage tracing, we demonstrate transdifferentiation of the cholangiocytes to hepatocytes. By comparing the transcriptomics of adult and larval liver, we find the presence of pro-plasticity mTORC1 signaling pathway in the uninjured liver of growing animal, which we validate using pharmacological inhibition. Overall, our findings show that the cellular mode of liver regeneration differs based on the stage and growth status of the organism.

## Results

### A zebrafish model of growth spurt
In zebrafish, the maternal yolk sustains the animal until 5 days post-fertilization (dpf), after which the animals are transferred to a diet of live rotifers. The nutrition supplied by the live food boosts the growth of the animal[26,27]. To evaluate the liver growth, we quantified the number of hepatocytes in the left lobe of zebrafish liver under normal rearing conditions (Supplementary Fig. 1). Hepatocytes were counted by performing in vivo confocal imaging of Tg(fabp10a:H2B-mGreen-Lantern) [abbreviated as Tg(fabp10a:H2B-mGL)] reporter line, which marks hepatocyte nuclei with green fluorescence (Supplementary Fig. 1A). From 4 to 6 dpf, there is no significant increase in hepatocyte number. However, we observed a 96% increase in hepatocyte numbers from 6 dpf to 8 dpf, and a 121% increase from 8 dpf to 11 dpf (Supplementary Fig. 1B). This demonstrates a rapid increase in hepatocytes from 6 – 11 dpf, representing the late larval stage. We selected this stage to evaluate the cellular source(s) of hepatocyte regeneration.

### CellCousin: a non-invasive partial genetic ablation model allowing segregation of spared and de novo hepatocytes
We developed a genetic system, named CellCousin, to investigate the cellular source of regeneration after partial ablation of hepatocytes. This system combines Cre/Lox-based stochastic genetic cell labeling, with Nitroreductase (NTR)-based cell ablation[28] (Fig. 1A). The Cre-driver line constitutes of the regulatory sequence of a pan-hepatocyte gene, fabp10a[29], driving tamoxifen-inducible CreER[T2]. The Cre-reporter line Tg(fabp10a:lox2272-loxp-nls-mTagBFP2-stop-lox2272-H2B-mGreenLantern-stop-loxp-mCherry-NTR), adapted from Brainbow construct[30], is also driven by the fabp10a promoter, and expresses nuclear mTagBFP2 in the default state. Upon Cre-activation using a pulse of 4-hydroxytamoxifen (4-OHT), the default mTagBFP2 signal is stochastically switched to either H2B-mGreenLantern (mGL)[31] or mCherry-NTR[32] (Fig. 1A, B). Subsequent treatment with Metronidazole (Mtz) ablates the mCherry-NTR expressing hepatocytes, while sparing the H2B-mGL-only expressing hepatocytes (Fig. 1B).

Moreover, after the partial ablation, the recovery of the hepatocytes could proceed in one of two ways (Fig. 1C): 1) By self-duplication of spared hepatocytes, whereby the H2B-mGL+ spared hepatocytes replenish the lost hepatocytes with H2B-mGL+ hepatocytes; or 2) By conversion of non-hepatocytes to hepatocytes, whereby cells that did not undergo Cre-based labeling would turn on the expression of the non-recombined Cre-reporter cassette and be labeled with the default nuclear mTagBFP2 expression. Thus, the double transgenic line (Fig. 1A), would allow the evaluation of spared vs de novo hepatocyte contribution to regeneration. As the spared and de novo hepatocytes represent the developmental and regenerative lineages, both originating from hepatoblasts during embryonic development, we term them as cellular cousins and name the technique as CellCousin.

We generated the Tg(fabp10a:CellCousin) line and evaluated its performance in late larval stage (9 dpf). Without 4-OHT treatment, we observed background recombination in 19.6 ± 11.2% of hepatocytes (Supplementary Fig. 2A, B). This is similar to the published background recombination frequency for fabp10a-driven CreER[T2] lines[16], potentially due to the strength of the transgene. Upon 4-OHT treatment from 4.5−5.5 dpf, the default blue fluorescence (nls-mTagBFP2) persists for about 3 days, with mCherry-NTR fluorescence taking approximately 4 days to become discernible (Supplementary Fig. 2C). At 10 dpf, we observed a complete conversion of mTagBFP2 expression to H2B-mGL (54.5 ± 14.4%) or mCherry-NTR (11.5 ± 15.1%) (Supplementary Fig. 2A, B). We also observed mGL+ / mCherry+ double positive cells (33.8 ± 20.5%) due to the insertion of multiple copies of the construct in the genome, as is common for meganuclease-based transgenesis[33]. In total, 45.3% of hepatocytes express mCherry-NTR after 4-OHT pulse, which represents the percentage of hepatocytes ablated using the CellCousin system.

### De novo hepatocytes emerge following partial ablation in larval zebrafish
Next, we established the protocol for partial ablation and subsequent evaluation of the source of regenerating hepatocytes. To accomplish ablation at the late larval stage in zebrafish, recombination was induced from 4.5 – 5.5 dpf, followed by ablation of mCherry-NTR expressing hepatocytes by treatment with 10 mM Mtz starting at 10.5 dpf (Fig. 1D). The treatment was conducted for two overnights, with regular feeding during the day. Longitudinal live imaging was performed on the left lobe of the liver of same animal at 0, 1, and 2 days post-partial ablation (dppa) (Fig. 1E). At 0 dppa, immediately after the end of the Mtz treatment, we observed aggregated mCherry signal, indicative of cell debris, while H2B-mGL+ hepatocytes were present scattered through the liver. At 1 dppa, we observed the emergence of mTagBFP2+ hepatocytes, which were previously absent in the same liver (Fig. 1E). The mTagBFP2+ population represents the de novo hepatocytes, which increased significantly at 1 and 5 dppa (Fig. 1F, G). These results demonstrate the robust contribution of de novo hepatocytes to the regenerative response following partial ablation in late larval stage zebrafish.

### De novo hepatocytes arise from transdifferentiation of cholangiocytes
To test if cholangiocytes are the origin of de novo hepatocytes in our partial ablation model, we performed short-term lineage tracing using the perdurance of histone-labeled fluorescent protein (Fig. 2A, Supplementary Fig. 3). Here, the reporter line, Tg(tp1:H2B-mCherry), marks the Notch-responsive cholangiocytes with H2B-mCherry[12,34–39]. Upon transdifferentiation of cholangiocytes, H2B-mCherry signal persists due to the low turnover of histones. The signal is progressively diluted by cell cycle and protein turnover; however, it provides a short window for tracing the cholangiocyte lineage. With this logic, the Tg(tp1:H2B-mCherry); Tg(fabp10a:CellCousin) line would label the early de novo hepatocytes with both H2B-mCherry and nls-mTagBFP2 (Fig. 2A). In contrast, late de novo hepatocytes, with further dilution of H2B-mCherry, are only nls-mTagBFP2 + .

In the vast majority of unablated animals, we did not observe any H2B-mCherry+ / nls-mTagBFP2+ cells (Supplementary Fig. 3).

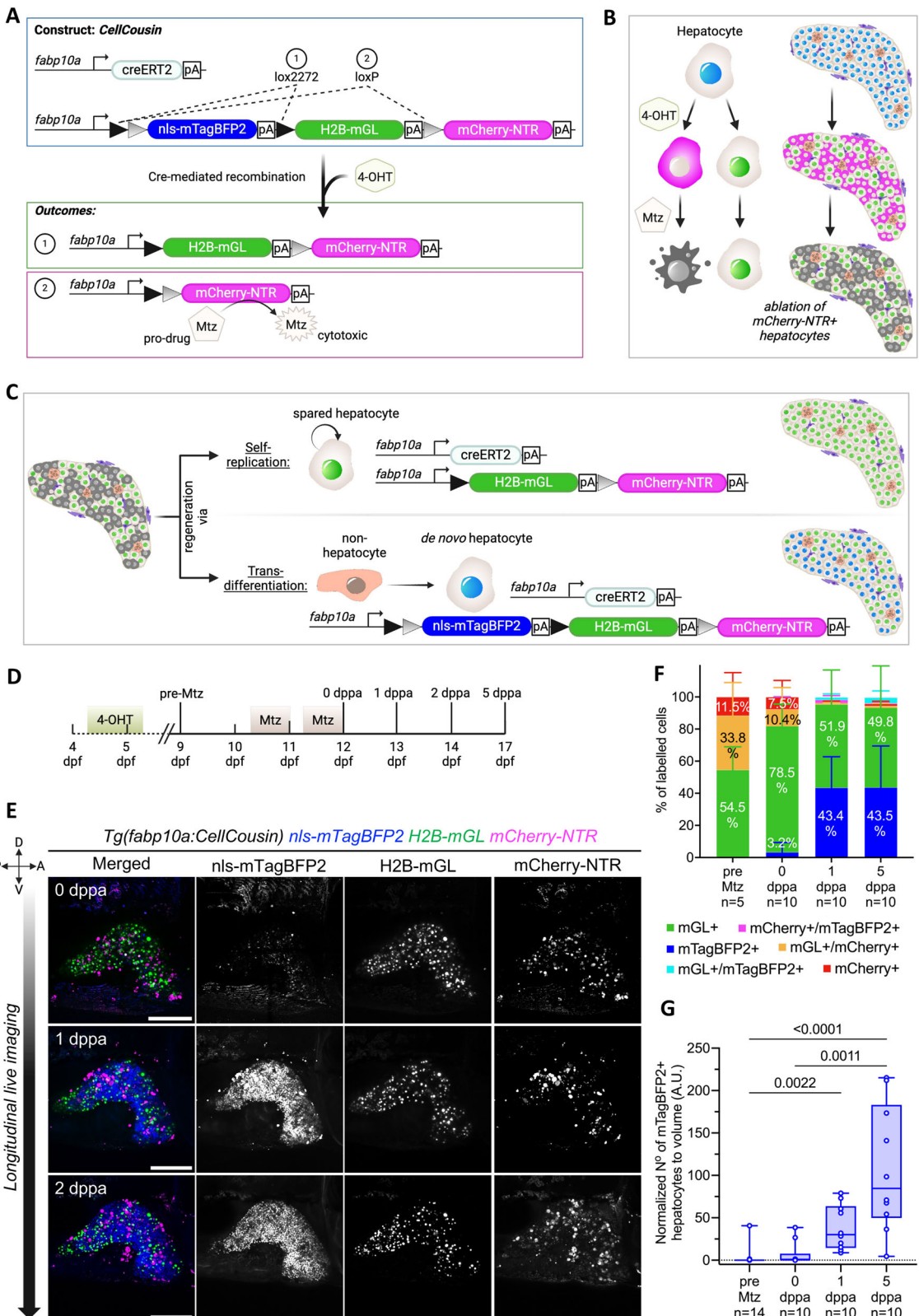

**Fig. 1 | CellCousin-enabled partial ablation of hepatocytes and subsequent tracking of hepatocyte regeneration. A** Schematic illustrating CellCousin construct for labeling hepatocytes by Cre/lox-mediated-recombination. **B** Strategy for the ablation of mCherry-NTR-expressing hepatocytes. **C** Schematic illustration showing the two potential modes of hepatocyte regeneration after partial ablation and the associated fluorescent labeling. **D** Schematic illustrating the experimental strategy. **E** Longitudinal in vivo confocal imaging of the same liver at 0, 1 and 2 dppa showing emergence of mTagBFP2+ hepatocytes in presence of spared H2B-mGL+ hepatocytes. Scale bar: 200 μm. **F** Barplots with mean ± SD of the percentage of hepatocyte labeling before (pre-Mtz, $n = 5$ animals) and after Mtz treatment (0,1 and 5 dppa, $n = 10$ animals each). **G** Min-to-max boxplot showing quantification of normalized number of mTagBFP2+ cells before and after Mtz treatment (pre-Mtz, $n = 14$ animals; after Mtz treatment (0,1 and 5 dppa), $n = 10$ animals each) (Kruskal Wallis test).

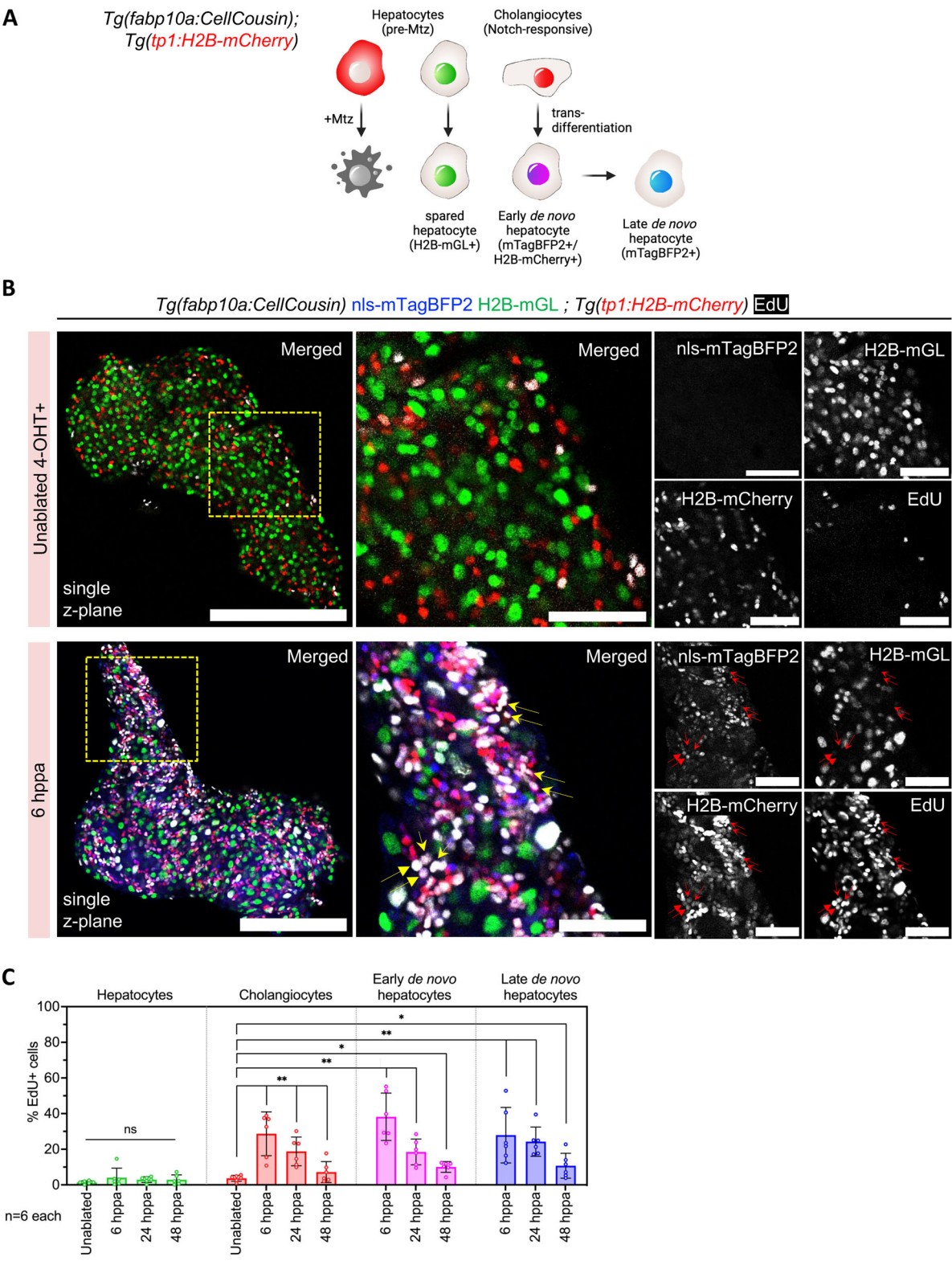

**C**

However, in all animals that underwent ablation, we observed the presence of H2B-mCherry+ / nls-mTagBFP2+ cells (Supplementary Fig. 3), indicating cholangiocytes as the source of de novo hepatocytes.

To further characterize the cellular response following partial ablation, we conducted a cell-cycle analysis using 5-Ethynyl-2′-deoxy-yuridine (EdU) incorporation assay and whole-mount imaging from 6 to 48 h post-partial ablation (hppa) (Fig. 2B, C). The analysis

demonstrated that spared hepatocytes show no significant increase in proliferation following partial ablation (Fig. 2C). In contrast, cholangiocytes display a robust increase in proliferation between 6 and 48 hppa, with $29.0 \pm 5.0\%$ of cholangiocytes being EdU+ at 6 hppa. Additionally, de novo hepatocytes, which are derived from cholangiocytes, show marked proliferation within this same timeframe (Fig. 2C). These findings support the conclusion that cholangiocytes,

**Fig. 2 | Cholangiocytes proliferate in response to partial ablation of hepatocytes. A** Schematic illustrating the lineage tracing of cholangiocytes after partial ablation via the CellCousin method. Hepatocytes are labeled with H2B-mGL+ or mCherry-NTR+ after 4-OHT treatment in the *Tg(fabp10a:CellCousin)* line. Cholangiocytes, which are Notch-responsive, are marked with bright H2B-mCherry+ (nuclear red) using the *Tg(tp1:H2B-mCherry)* line. After MTZ administration, Specific ablation of mCherry-NTR+ hepatocytes leads to a liver with spared H2B-mGL+ hepatocytes and de novo nls-mTagBFP2+ hepatocytes. Early de novo hepatocytes derived from cholangiocytes are detected as nls-mTagBFP2 + /H2B-mCherry+ (nuclear magenta), while late de novo hepatocytes, due to dilution of H2B-mCherry label, are marked only with nls-mTagBFP2. **B** Confocal images of livers from EdU labeled uninjured and 6 hppa *Tg(fabp10a:CellCousin); Tg(tp1:H2B-mCherry)* transgenic animals. Arrows mark proliferating early de novo hepatocytes. Scale bar: 200 μm (left panels), 50 μm (insets). **C** Barplot with Mean ± SD showing quantification of the percentage of EdU+ hepatocytes, cholangiocytes, and early and late de novo hepatocytes in uninjured, 6 dppa, 24 dppa and 48 dppa (*n* = 6 animals). (ANOVA, followed by pair wise comparison using Two-Tailed Mann Whitney. For cholangiocytes and de novo hepatocytes, comparison with uninjured cholangiocytes is shown. ** *p*-value = 0.0022; * *p*-value < 0.05).

rather than spared hepatocytes, serve as the primary source of new hepatocytes in this regeneration model.

## Establishment of a partial hepatectomy (PHx) model in larval zebrafish

Partial hepatectomy (PHx), the surgical removal of a portion of the liver, is a widely used model for studying liver regeneration due to its ability to mimic clinically relevant liver injuries. The procedure results in a regionally restricted injury in which the hepatocytes as well as non-parenchymal cells, including cholangiocytes, are removed. In response, a coordinated tissue restoration program needs to be activated for successful regeneration.

Here, we established PHx injury model for late larval stage zebrafish (10 dpf, standard length (SL) of ~ 5 mm) (Fig. 3A). At 10 dpf, the liver is composed of two lateral lobes, the left and the right lobe, connected in the anterior by a semi-circular ring. For PHx, we resected the centre of the left lobe and removed the region posterior to the cut site (Fig. 3A, B). Using *Tg(fapb10a:H2B-mGL)* we quantified the amount of lost tissue and observed a reduction of 17.3% (uninjured 9901 ± 1631, injured 8184 ± 1311) in hepatocytes upon PHx (Fig. 3C).

Following PHx, we performed cell-cycle analysis at 1-, 2- and 3-days post-injury (dpi) using whole-mount EdU imaging to assess the proliferative response. For this, we utilized *Tg(fabp10a:H2B-mGL); Tg(tp1:H2B-mCherry)* transgenic line, in which hepatocytes and cholangiocytes are marked with nuclear green and red fluorescence, respectively. Strikingly, we found that EdU+ cells were localized within 50 μm of the regenerate edge at 1 dpi, indicating the formation of a high-proliferation zone immediately following injury (Fig. 3D; Supplementary Movie 1). Thus, we divided the left and right lobe into two regions: a distal region of 50 μm width, and a proximal region encompassing the remaining part of the lobe (Fig. 3E). Within the distal region of the injured left lobe, hepatocytes exhibit a significant increase in cell-cycle activity at 1 dpi (Fig. 3F), while cholangiocytes showed robust cell-cycle activity at both 1 and 2 dpi (Fig. 3G). Notably, 20.9 ± 2.1% of cholangiocytes within 50 μm of the regenerate edge enter S-phase at 1 dpi (Fig. 3G), highlighting a strong localized proliferative response. Outside of this zone, neither hepatocytes nor cholangiocytes display increased cell-cycle activity, indicating that the regenerative response remains concentrated at the injury site. Furthermore, cholangiocytes sustain their proliferative response longer and more robustly than hepatocytes, suggesting a regionally restricted role for cholangiocytes in supporting liver regeneration.

Next, we evaluated the recovery of the liver after PHx by comparing the length of the resected left lobe with the unharmed right lobe at 4 dpi, 11 dpi and at 4 months post-injury (mpi) (Supplementary Fig. 4A). Additionally, we measured the length of the ventral lobe (Supplementary Fig. 4B), which starts developing as an outgrowth from the left lobe along the ventral midline around 18–20 dpf[40]. At both 4 and 11 dpi, the ratio of the left to right lobe was significantly shorter in animals undergoing PHx than in controls (Supplementary Fig. 4C–F). Moreover, at 11 dpi, the ventral lobe was shorter in PHx condition as compared to controls (Supplementary Fig. 4E, F). At 4 mpi, which represents the adult stage, the ratio of left to right lobe was not different between the PHx and control animals (Supplementary

Fig. 4G, H). Intriguingly, the ventral lobe remained shorter in the adults that underwent PHx as compared to controls (Supplementary Fig. 4G, H), suggesting that PHx has a long-term impact on the morphology of the liver in zebrafish. Further, sham surgery did not impact ventral lobe length or liver volume at 11 dpi (Supplementary Fig. 4I, J). However, since specific markers to label the ventral lobe anlage are currently unavailable in zebrafish, we are unable to track the precise origins or status of this anlage post-surgery. It is therefore possible that the ventral lobe anlage is directly impacted or even partially damaged during the left lobe PHx, which could account for its diminished growth.

## De novo hepatocytes emerge after partial hepatectomy in late larval stage zebrafish

Next, we investigated the source of hepatocyte regeneration in the newly established PHx model. To this end, we utilized fate mapping of hepatocytes that were spared from the injury to evaluate their contribution to hepatocyte recovery (Fig. 4A). For this, we labeled all the hepatocytes using the CellCousin system by performing 4-OHT-based Cre recombination from 4.5–5.5 dpf (Fig. 4B). As previously demonstrated (Supplementary Fig. 2A, B), the recombination event leads to the exchange of default nuclear-mTagBFP2 fluorescence with H2B-mGL or mCherry fluorescence in the entire hepatocyte population. At 10 dpf, six days after the recombination event, we performed PHx. Following PHx, hepatocytes derived from the spared, pre-existing hepatocytes would be labeled with H2B-mGL or mCherry (Fig. 4A). However, any hepatocytes derived from non-hepatocytes would harbor the nuclear-mTagBFP2 mark, due to the lack of Cre activation in such cells at the time of 4-OHT treatment (Fig. 4A). At 4 dpi, we observed significantly more mTagBFP2+ hepatocytes in PHx condition as compared to controls (Fig. 4C–E), demonstrating the emergence of de novo hepatocytes. This indicates that hepatocytes originate from non-hepatocyte source(s) in response to injury.

## De novo hepatocytes display a temporal cholangiocyte-to-hepatocyte molecular transition

To explore the cellular and molecular landscape during regeneration after partial ablation and hepatectomy, we conducted single-cell RNA-Sequencing (scRNA-seq) using the 10X Genomics platform of larval zebrafish liver (12 to 21 dpf) isolated from *Tg(fabp10a:CellCousin)* fish. This was performed in the basal state (*N* = 2), after partial ablation (0,1 and 9 dppa) and after partial hepatectomy (4 and 11 dpi) (Fig. 5A).

After regular quality control, we obtained a total of 30,856 cells for analysis (Fig. 5B, Supplementary Fig. 5A). Unsupervised clustering of the single-cell transcriptome data identified 10 clusters, representing hepatocytes (51.7%), cholangiocytes (6.2%), endothelial cells (13.1%), hepatic stellate cells (HSCs) (21.0%), macrophages (4.3%), neutrophils (0.4%), lymphocytes (1.1%), erythrocytes (0.8%), and small contamination of acinar cells (1.0%) and intestinal cells (0.3%) (Fig. 5B, C).

Next, we marked mTagBFP2+ cells based on the detection of mTagBFP2 mRNA in the single-cell data within the cholangiocyte and hepatocyte cluster. With that, we labeled hepatocytes under control condition (4381 cells), cholangiocytes (704 cells in control and 649

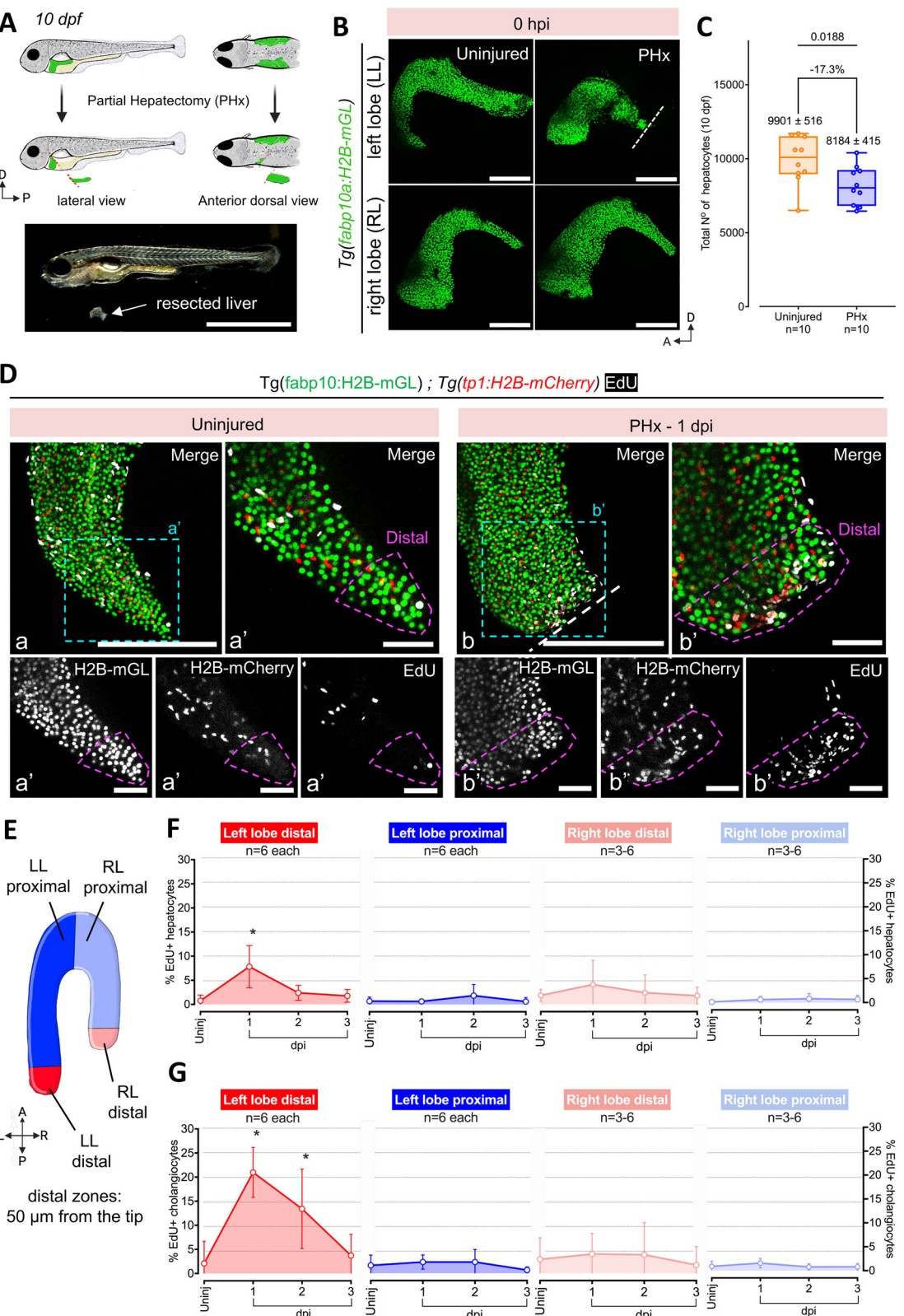

**Fig. 3 | Establishment of a partial hepatectomy (PHx) model of liver regeneration. A** Schematic illustration (above) and brightfield image (below) showing the surgical procedure for PHx of the left lobe in 10 dpf larvae. **B** Representative confocal images of the left and right liver lobe from uninjured or 0 hours-post-injury (hpi) PHx *Tg(fabp10a:H2B-mGL)* animals. Site of resection marked with white dashed line. **C** Min-to-max boxplot showing quantification of the total number of hepatocytes (nuclei) at 10 dpf in the livers subjected to PHx (*n* = 10 animals, each) (Two-tailed Welch's t test). Each dot represents one animal; mean ± SEM is indicated above each box. **D** Confocal images of livers from EdU labeled

*Tg(fabp10a:H2B-mGL); Tg(tp1:H2B-mCherry)* animals at 11 dpf (uninjured) and 1 dpi. Scale bar: 200 μm (a, b), 50 μm (a', b'). **E** Schematic depicting the segregation of the liver lobes into a distal region of 50 μm thickness and a proximal region. **F, G** Quantification representing Mean ± SD of the percentage of EdU+ hepatocytes (**F**) and cholangiocytes (**G**) in uninjured, 1-, 2- and 3-dpi separated by distal and proximal regions of the left (injured) (*n* = 6 animals each) and right (uninjured) (*n* = 6, 6, 3, 5 animals, respectively) lobe. (ANOVA followed by Kruskal–Wallis test; *p-value < 0.05).

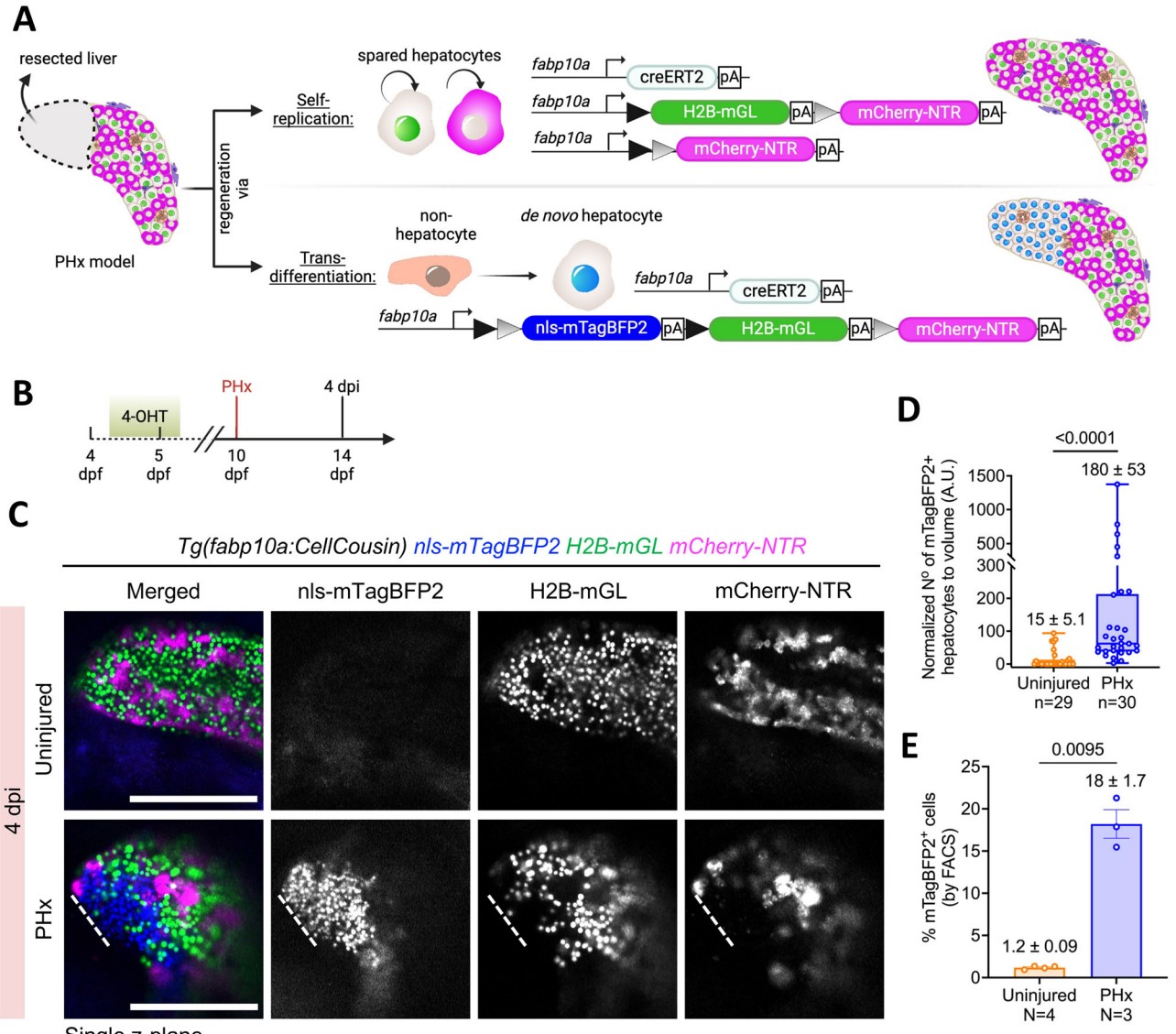

**Fig. 4 | Contribution of de novo hepatocytes to regeneration following partial hepatectomy (PHx). A** Schematic showing the potential modes of regeneration after PHx and the associated fluorescent labeling of hepatocytes. **B** Experimental strategy for 4-OHT-pulse labeling and PHx in the *Tg(fabp10a:CellCousin)* background. **C** Representative confocal images of the left lobe from live imaging of the *Tg(fabp10a:CellCousin)* animals that were uninjured or subjected to PHx. **D** Min-to-max boxplot showing quantification of the normalized number of mTagBFP2+ cells

in uninjured ($n = 29$ animals) and PHx animals ($n = 30$ animals) at 4 dpi (Two-tailed Mann–Whitney test). Each dot represents one animal; mean ± SEM is indicated above each box. Scale bar: 200 μm (**B**). **E** Barplot with Mean ± SEM showing quantification of the percentage of mTagBFP2+ hepatocytes using FACS in uninjured ($n = 4$ biological replicates) and 4 dpi ($n = 3$ biological replicates) animals (Two-tailed Welch's t-test). Each dot represents one biological replicate; mean ± SEM is indicated above each box.

cells in injured conditions), de novo hepatocytes (mTagBFP2+ cells 6499 cells) and spared hepatocytes (mTagBFP2- hepatocytes 5631 cells) as per condition (Fig. 5D, Supplementary Fig. 5B).

After cell annotations, we evaluated the transcriptomic similarity between mTagBFP2+ cells and hepatocytes & cholangiocytes from the uninjured condition. Correlation analysis and hierarchical clustering revealed that mTagBFP2+ cells at 0 dppa, 1 dppa and at 4 dpi were similar to uninjured cholangiocytes (Fig. 5E). In contrast, at 9 dppa and 11 dpi, mTagBFP2+ cells display convergence with spared and uninjured hepatocytes. Moreover, mTagBFP2+ cells exhibit a temporal decrease in the expression of cholangiocyte-specific marker genes, including *anxa4* and *epcam*, and a corresponding increase in the expression of hepatocyte-specific marker genes (Fig. 5F and Supplementary Fig. 5C).

To determine which specific cholangiocyte populations have the potential to give rise to hepatocytes, we performed pseudotime

trajectory analysis on cholangiocyte subclusters. Sub-clustering of control scRNA-seq data identified three distinct cholangiocyte subpopulations in the uninjured liver: luminal cholangiocytes, marked by *krt4* expression[41]; a proliferative cholangiocyte cluster; and a mixed cluster containing both intrahepatic (IHD) and intermediate cholangiocytes, which express *her2*, *her9*, and *her6* (Supplementary Fig. 6A–E)[41]. These subpopulations align with the distinct maturation stages of zebrafish cholangiocytes, following a differentiation trajectory from IHD to intermediate to luminal, where luminal cholangiocytes represent the most mature cholangiocyte population[41].

To further investigate cholangiocyte differentiation dynamics, we performed pseudotime trajectory analysis, incorporating both cholangiocyte subclusters and hepatocytes. The inferred trajectory revealed that IHD and intermediate cholangiocytes serve as the primary source of potential transdifferentiation events (Supplementary Fig. 6F, G). Along the pseudotime path, we observed a progressive shift

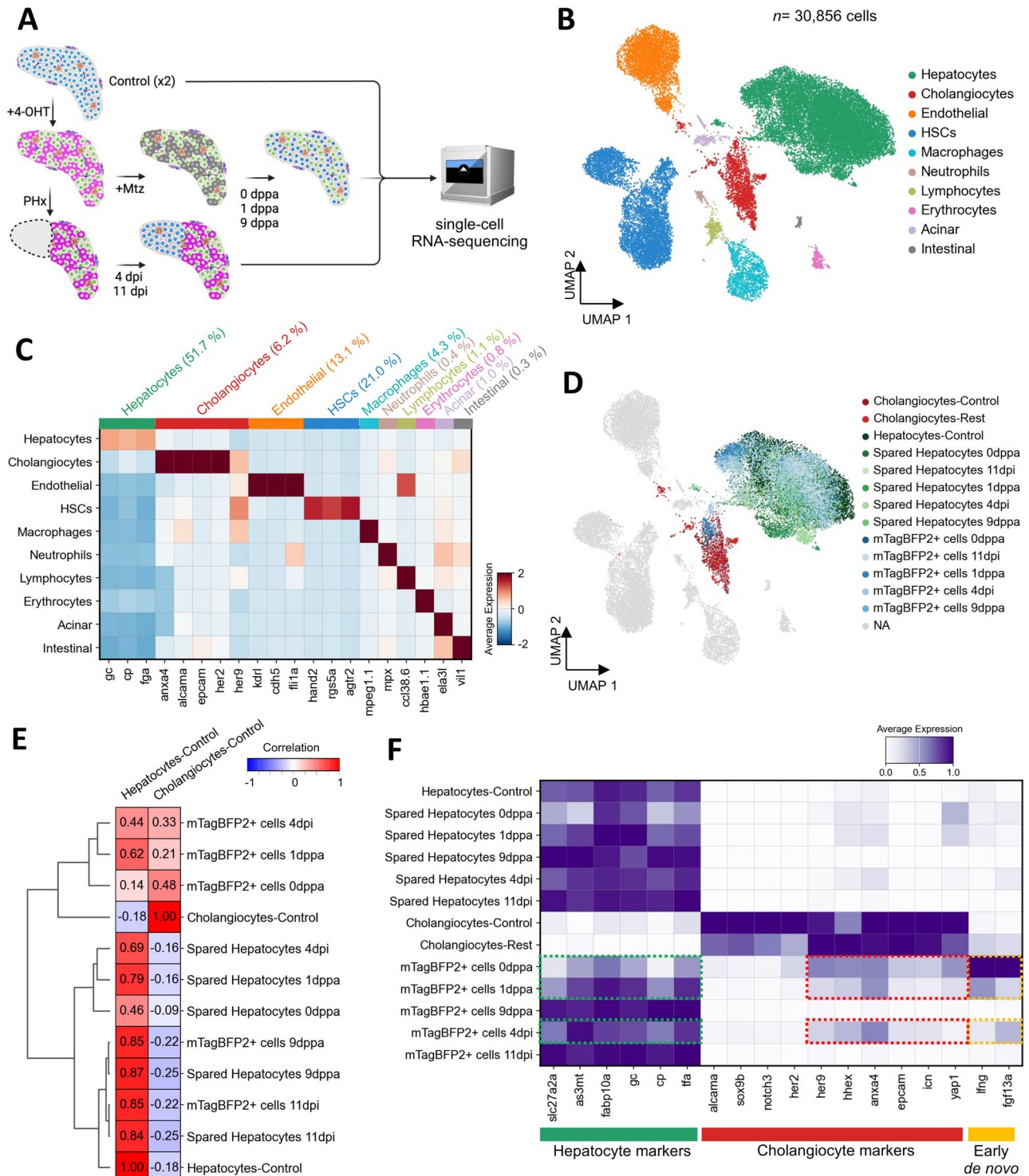

**Fig. 5 | De novo hepatocytes display transcriptional convergence with uninjured hepatocytes. A** Schematic representation of the single-cell RNA-seq approach to investigate the cellular and transcriptional landscape in basal condition (control-uninjured, $N = 2$) and during regeneration after partial ablation (0,1,9 dppa) and partial hepatectomy (4, 11 dpi). **B** UMAP visualization of cell types detected in zebrafish larval liver, combining data from injured and uninjured liver conditions. **C** Heatmap displaying the average scaled expression of selected markers across different cell types. **D** UMAP visualization focusing on specific injured conditions, highlighting mTagBFP2+ cells and spared hepatocytes (0,1,9 dppa and 4,11 dpi), control hepatocytes, control cholangiocytes, and cholangiocytes in injured conditions (cholangiocytes-rest). The remaining clusters are shown in gray. **E** Correlation matrix of hierarchically clustered mTagBFP2+ and spared hepatocytes compared to control hepatocytes and cholangiocytes. **F** Heatmap showing the normalized expression of hepatocyte and cholangiocyte-specific marker genes. Expression of selected genes that are detected in mTagBFP2+ cells at 0 dppa, 1 dppa and 4 dpi are outlined (dashed rectangles).

in gene expression, where her2 expression gradually decreased, while fabp10a expression became enriched, marking the transition toward a hepatocyte-like fate (Supplementary Fig. 6H).

Together, this data indicates that cholangiocytes are the source of de novo hepatocytes in response to injury, and that these cells converge with hepatocytes at transcriptomic level at late stages of regeneration.

De novo hepatocytes at the early stages of maturation specifically expressed *lfng* (lunatic fringe) and *fgf13a* (Fig. 3F). As *lfng* is a Notch inhibitor[42,43], we wondered if we could induce cholangiocyte-to-hepatocyte transdifferentiation without injury by inhibiting Notch signaling. To this end, we treated 4-OHT labeled *Tg(fabp10a:CellCousin)* larvae with LY411575, a gamma-secretase inhibitor (Supplementary Fig. 7A). Notably, no Mtz treatment was performed. We observed that Notch inhibition led to the presence of mTagBFP2+ cells, even in the absence of ablation (Supplementary Fig. 7B, C). Thus, inhibiting Notch signaling can lead to spontaneous emergence of de novo hepatocytes.

### Lineage tracing validates cholangiocyte contribution to hepatocytes after PHx

Our scRNA-seq data revealed that de novo hepatocytes exhibit cholangiocyte characteristics in their transcriptomic profiles at early stages of regeneration (Fig. 5E, F). To conclusively demonstrate that cholangiocytes represent the origin of the de novo hepatocytes following PHx, we performed lineage tracing of the cholangiocyte population. To label the cholangiocytes, we generated Cre-driver lines by knocking-in CreER[T2] in the endogenous locus, thereby avoiding any potential positional effects or transgene silencing related to random transgenesis.

To this end, we utilized our scRNA-seq dataset to identify marker genes enriched in the cholangiocyte population in the uninjured condition. We focused on transcriptional targets of Notch signaling as Notch is the major regulator of cholangiocyte identity (Supplementary Fig. 7). Here, we identified *her2* and *her9* as potential candidates for generation of Cre-driver lines. In the scRNA-seq dataset, *her2* is specifically expressed in the cholangiocytes, whereas *her9* is also expressed in the non-parenchymal cells (Fig. 5C). Importantly, the expression of both genes is excluded from hepatocytes. For both genes the Cre-drive lines were generated by inserting the p2A-EGFP-t2A-CreER[T2] sequence immediately upstream the stop codon[44]. The viral 2A sequences lead to cleavage of CreER[T2] protein from the gene product, which would minimize the impact of the knock-in on the protein function.

Next, the her2 tracer line, *TgKI(her2-p2A-EGFP-t2A-CreER[T2])* [abbreviated as TgKI(her2-CreER[T2])] was crossed to a Cre-responder line, *Tg(ubi:loxp-CFP-stop-loxp-H2B-mCherry)* [abbreviated as *Tg(ubi:CSHm)*] in the background of a hepatocyte reporter line, *Tg(fabp10a:H2B-mGL)* (Fig. 6A). In this setup, the Cre recombination event results in the switch from cyan fluorescent protein (CFP) to histone-tagged nuclear mCherry fluorescence in the her2-expressing cells. As the hepatocytes are marked with nuclear mGL signal, hepatocytes derived from the her2-expressing cells will be represented by the nuclear mCherry+ / mGL+ signal (Fig. 6B).

The her2+ cells were labeled with a pulse of 4-OHT treatment from 2 – 3.5 dpf (Fig. 6C). To ensure that cholangiocytes were labeled with the *TgKI(her2-CreER[T2])* driver line, we stained the livers with the antibody against cholangiocyte-specific *Anxa4* gene (using the 2F11 antibody)[45]. We observed that the her2 Cre driver line labels 80.4% of cholangiocytes in the liver (Supplementary Fig. 8A, B). Further, among the her2-lineage cells, marked with mCherry fluorescence, 93.7% were identified as cholangiocytes (Supplementary Fig. 8C), indicating that the her2 lineage is predominantly specific to cholangiocytes in the liver.

Subsequently, PHx was performed at 10 dpf in *TgKI(her2-CreER[T2]); Tg(ubi:CSHm); Tg(fabp10a:H2B-mGL)* triple transgenic line, seven days after Cre-based labeling, and samples collected at 4 and 8 dpi. At both the time points sampled, the liver in the PHx animals demonstrated a significantly higher number of mCherry+ / mGL+ cells as compared to controls (Fig. 6D–G). This demonstrates that the cholangiocytes are the source of de novo hepatocytes after PHx.

We similarly repeated the analysis with her9 lineage tracing line. For the her9 Cre driver line, it labeled 59.3% of cholangiocytes (Supplementary Fig. 9A–C). Within the her9 lineage, 60.2% were cholangiocytes (Supplementary Fig. 9D), demonstrating that cholangiocytes are part of the her9+ cell lineage. Additionally, our single-cell RNA-Seq dataset revealed that *her9* is expressed in *hand2+* cells (Fig. 5C). Using the *TgBAC(hand2:EGFP)* line, which labels hepatic stellate cells, mesothelial cells and perivascular cells[46], we identified that 30.0% of her9-lineage cells were hand2+ (Supplementary Fig. 9D, E). For lineage tracing after PHx, 4-OHT labeling of *TgKI(her9-CreER[T2]); Tg(ubb:CSHm); Tg(fabap10a:H2B-mGL)* was conducted from 2 – 3.5 dpf (Supplementary Fig. 10A–C). Analysis of her9+ cell lineage at 4 and 8 dpi displayed significantly more mCherry+ / mGL+ cells in the livers that underwent PHx as compared to uninjured controls (Supplementary Fig. 10D–G), similar to the *her2+* cell lineage tracing. This further validates that non-hepatocytes can generate hepatocytes after PHx.

### mTORC1 signaling regulates cholangicyte plasticity in larval zebrafish

As partial hepatectomy at the adult stage does not induce transdifferentiation of Notch-responsive cholangiocytes in zebrafish[9], we aimed to investigate potential differences in cholangiocytes between larval and adult zebrafish liver. To identify the molecular differences, we examined the transcriptional profiles of liver cells from uninjured control 13 dpf larval liver and 18 months post-fertilization (mpf) adult liver. By integrating our uninjured control larval dataset with the public adult dataset[47], observed that the transcriptional profiles of different cell types in the larval and adult liver are broadly conserved (Fig. 7A, B), underscoring the fact that the larval liver does not contain a distinct cholangiocyte population. However, differential gene expression analysis comparing larval cholangiocytes ($n = 704$ cells) to adult cholangiocytes ($n = 1,106$ cells) highlighted significant differences in signaling pathways (Supplementary Data 1). Notably, there was a downregulation of *sox9b* expression in adult stage cholangiocytes compared to larval cholangiocytes (Fig. 7C), which is a key gene responsible for the plasticity of the cholangiocytes[12,48–51].

Furthermore, larval cholangiocytes exhibited upregulation of genes associated with the mTORC1 signaling pathway (Fig. 7C), which has been reported to be critical for the transdifferentiation of cholangiocytes[52–54]. We also performed differential gene expression analysis of cholangiocyte subclusters (Supplementary Fig. 11A–C). Differentially expressed genes between larval and adult stage corresponding to the IHD/intermediate and luminal cholangiocytes are provided in Supplementary Data 2 and Supplementary Data 3, respectively. This analysis revealed that mTORC1 pathway activity-related genes are elevated in both IHD/intermediate and luminal cholangiocytes from larval livers compared to adults (Supplementary Fig. 11D, E).

To validate the differential activity of the mTORC1 pathway between larval and adult zebrafish in uninjured context, we performed staining for the phosphorylated ribosomal S6 protein (pS6), a downstream effector of mTORC1, in *Tg(fabp10a:H2B-mGL); Tg(tp1:H2B-mCherry)* animals. We observed that hepatocytes (indicated by nuclear mGL) and cholangiocytes (indicated by nuclear mCherry) were pS6+ in larval zebrafish liver (Fig. 7D, E). Conversely, pS6 staining was not detected in hepatocytes or cholangiocytes at the adult stage by immunofluorescence (Fig. 7D, E), in line with protein levels detected by Western blotting (Fig. 7F), indicating a reduced activity of the mTORC1 pathway in uninjured liver of adult zebrafish as compared to 10 dpf larval zebrafish.

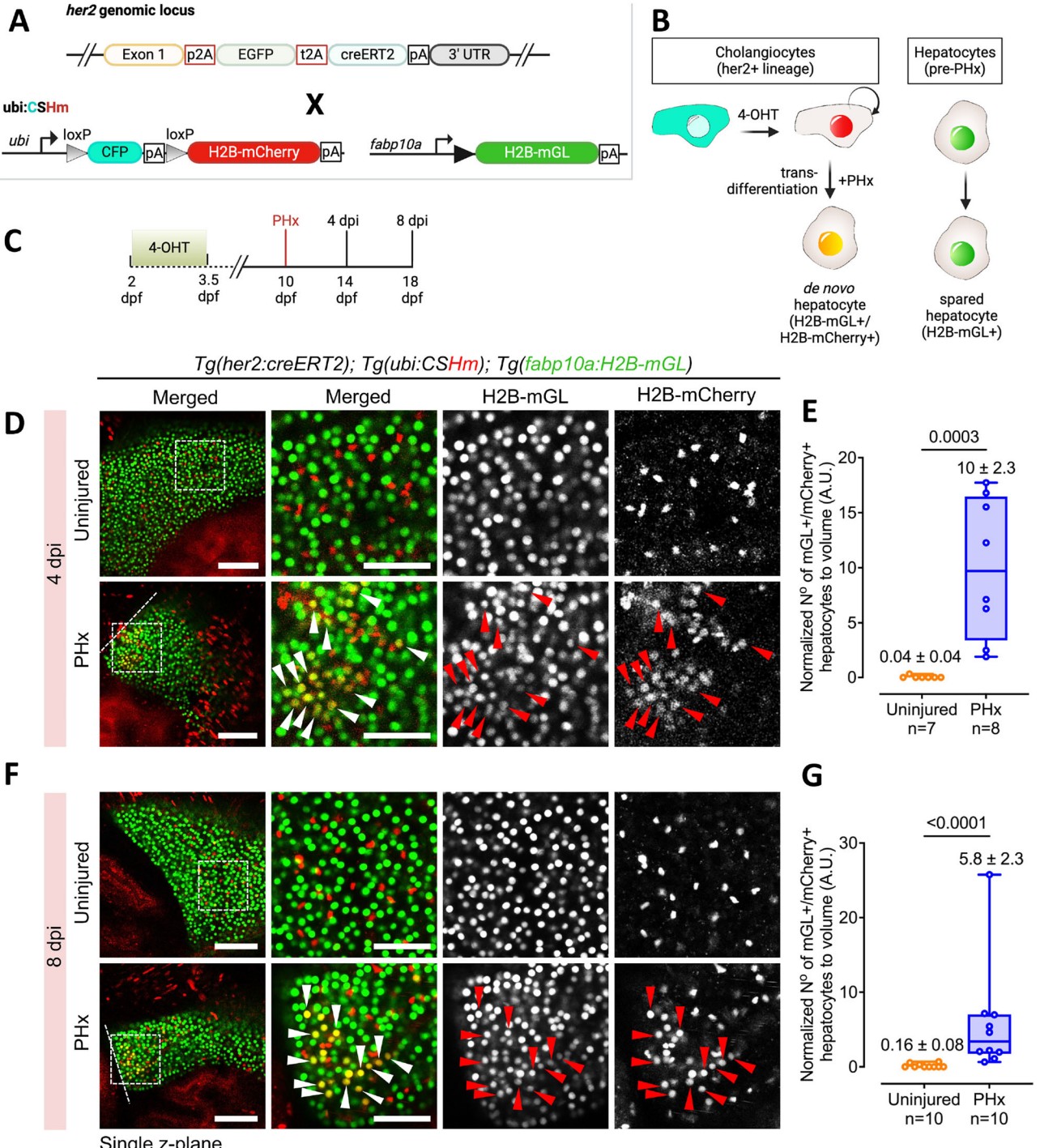

**Fig. 6 | her2+ cells generate hepatocytes after PHx. A** The transgenic lines used to lineage trace her2+ cells. **B** Schematic illustrating the tracing of her2 + -derived hepatocytes following PHx. **C** Experimental timeline for lineage tracing. **D**, **F** Live confocal images of left lobe in uninjured and PHx condition at 4 dpi (**D**) and at 8 dpi (**F**). Representative cholangiocyte-derived de novo hepatocytes are marked with arrowheads. Dashed line marks the border of the regenerate. **E**, **G** Min-to-max box plots showing the normalized number of H2B-mGL$^+$/mCherry$^+$ hepatocytes per unit volume at 4 dpi (**E**) and 8 dpi (**G**). Each dot represents one animal; mean ± SEM is indicated above each box. Statistical significance was assessed using the Two-tailed Mann–Whitney test. Sample sizes are indicated below the respective groups. Scale bars: 100 μm (left panels), 50 μm (insets) (**D**, **F**).

Next, we tested the requirement of the mTORC1 signaling pathway for cholangiocyte plasticity by inhibiting the pathway using rapamycin in the uninjured liver. Larvae at 12 dpf *Tg(fabp10a:CellCousin)* were subjected to overnight treatments with 10 μM rapamycin, while control animals were treated with DMSO (Fig. 7G). After treatment, the animals were subjected to partial hepatectomy at 14 dpf. It is crucial to note that post-injury treatments were avoided to

prevent any direct impact of the drugs on the transdifferentiation of cholangiocytes.

At 3 dpi, we observed significantly higher de novo hepatocytes in the DMSO-treated condition compared to the rapamycin-treated animals (Fig. 7H, I). This suggests that inhibition of the mTOR pathway in the uninjured liver can reduce the appearance of de novo hepatocytes post-injury, indicating that the mTORC1 signaling pathway

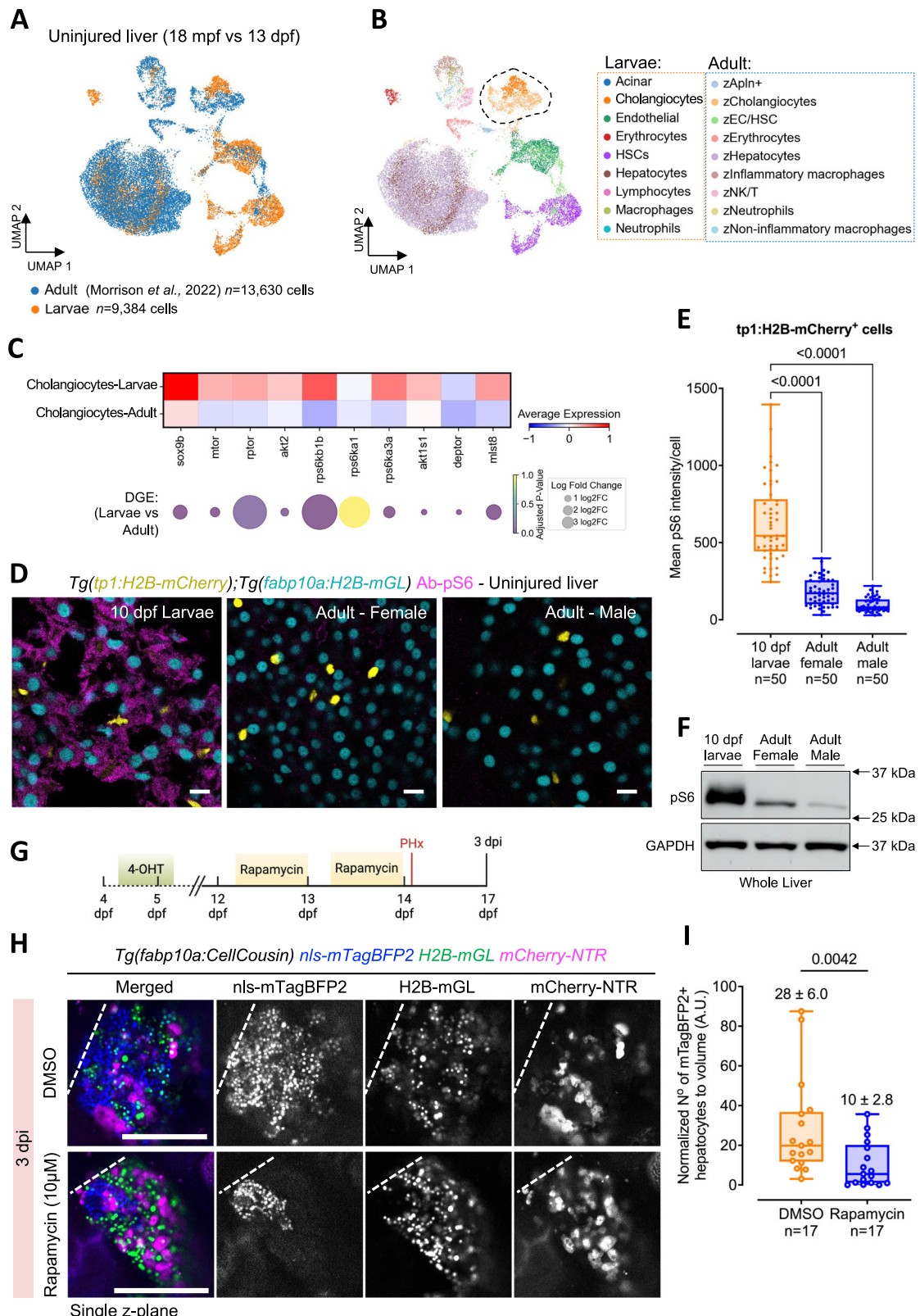

regulates the enhanced plasticity of cholangiocytes in the larval zebrafish.

## Discussion

Our observations challenge the current paradigm in liver regeneration, which posits that cholangiocytes contribute to the hepatocyte recovery process only under conditions of extensive injury or when hepatocytes are unable to enter the cell cycle[1,4–6,9–11]. Our findings demonstrate that cholangiocytes exhibit high plasticity and robustly contribute to hepatocyte regeneration even in the presence of spared hepatocytes following acute injury in late-stage larval zebrafish. Using single-cell RNA Sequencing, we show that the de novo hepatocytes co-express markers of cholangiocytes and hepatocytes at the early stages of regeneration (Fig. 5F). Similarly, co-expression of cholangiocyte and

**Fig. 7 | mTORC1 activity in uninjured liver regulates cholangiocyte plasticity during liver regeneration.** **A** UMAP visualization showing the integration of uninjured zebrafish larval and adult liver cells. **B** Unsupervised clustering of cell types presents in larval and adult datasets. The dashed area indicates cholangiocytes present in the two datasets. **C** Heatmap of mTORC1 signaling pathway-related gene expression in larval and adult cholangiocytes (top). Fold changes and *p*-values from differential gene expression (DGE) analysis displayed in dot plot (bottom). **D** Immunostaining for pS6 in uninjured larval and adult zebrafish liver sections. Scale bars: 10 μm. **E** Min-to-max boxplot showing quantification of pS6 intensity in cholangiocytes from livers of uninjured larval and adult zebrafish (*n* = 50 cells from 10 animals) (ANOVA followed by Kruskal Wallis test). **F** Western blot against pS6

and GAPDH for whole livers from 10 dpf larvae and adult zebrafish. **G** Schematic of the experimental approach for inhibiting the mTORC1 signaling pathway in *Tg(fabp10a:CellCousin)* zebrafish larvae. **H** Confocal images of the liver from 3 dpi *Tg(fabp10a:CellCousin)* larvae treated with DMSO or 10 μM Rapamycin. Dashed line marks the border of the regenerate. Scale bars: 200 μm. **I** Min-to-max boxplot showing quantification of normalized mTagBFP2+ de novo hepatocytes following treatment with Rapamycin treatment (*n* = 17 animals) compared to DMSO controls (*n* = 17 animals). Each dot represents one animal; mean ± SEM is indicated above each box. Statistical significance was assessed using the Two-tailed Mann–Whitney test.

hepatocyte marker genes was observed in liver biopsies from patients with chronic hepatitis and cirrhosis[18–21], as well in mouse models of severe liver injuries[11,13]. Thus, a transitional cellular stage is conserved between our model of partial injuries and the mammalian models of chronic liver diseases. Further, *fgf13a* in zebrafish (Fig. 5F) and FGF13 in humans[18] seems to be a marker of the transitional cellular stage.

It is important to highlight here that the cholangiocytes do not contribute to the hepatocyte population in the absence of an injury, as demonstrated by the lack of de novo hepatocytes in our control conditions (Figs. 1F, G and 6E, G). Thus, growth of the liver occurs predominantly by self-duplication of hepatocytes in zebrafish, which is in agreement with previous findings[2,5,9,16]. Moreover, we do not detect cells co-expressing hepatocyte and cholangiocyte markers in the scRNA-seq of uninjured liver (Fig. 5F), underscoring the lack of crossover between the two lineages without injury.

Within the cholangiocyte population, our pseudotemporal analysis revealed that intrahepatic (IHD) and intermediate cholangiocytes serve as the primary source of cholangiocyte-to-hepatocyte transdifferentiation, while luminal cholangiocytes, represented by krt4 expression[41], do not appear to contribute (Supplementary Fig. 6). Future studies will be needed to determine whether luminal cholangiocytes retain the capacity to alter their identity under specific conditions or if they are terminally differentiated.

In line with our work, a recent publication by Ambrosio et al. demonstrated that targeted local ablation of 10-20% of hepatocytes in 6 dpf zebrafish triggers cholangiocyte-mediated regeneration[55]. However, the authors proposed a changing ratio of hepatocytes to cholangiocytes after hepatocyte ablation as the trigger for ductular cell reaction. This hypothesis would not hold for PHx, where both hepatocytes and cholangiocytes are lost. Thus, our work shows that cholangiocyte contribution to hepatocytes occurs in a wider scope than previously reported.

We attribute the enhanced plasticity of the cholangiocytes to the stage of the animal at the time of injury. The liver of a growing animal could respond differently to injury as compared to the liver of an adult. In adults, hepatocytes are predominantly quiescent at homeostasis[5,22]. A subset of quiescent hepatocytes can enter cell-cycle to repopulate the liver, while the other hepatocytes continue to function, leading to a division of labor within the hepatocyte population[3,4]. However, in young animals, it is plausible that the demands of growth, function and regeneration cannot be managed by the hepatocytes, particularly when the organ size quadruples in a span of five days (Supplementary Fig. 1). Interestingly, despite partial ablation, spared hepatocytes do not increase their proliferation (Fig. 2C). Furthermore, after PHx, the edge of the regenerate displays a high level of proliferation in a regionally restricted manner (Fig. 3D–G), forming a "wound border zone." Within this border zone, the hepatocyte proliferation shows lower potentiation as compared to the cholangiocyte proliferation. These data suggest a potential bottleneck in the proliferative response of hepatocytes to the injury.

In addition to the proliferative differences, the difference between the liver at the postnatal and adult stage in mammals also extends to the organ's architecture. In mouse, the liver lobule, the characteristic

anatomic unit of the liver, is not present until postnatal day 7[56]. The lobules attain their final size and numbers at weaning, which occurs in the third week of birth[56]. Though a few attempts have been made to study liver regeneration in the first two weeks after birth in mouse[57,58], a detailed description of the cellular response to liver injury in early postnatal mouse is missing. In future, it would be of interest to compare liver regeneration in juvenile and adult mammals.

In our work, we show that the enhanced plasticity of cholangiocytes is regulated by mTORC1 pathway, which is active in uninjured livers during the growth spurt (Fig. 7D–F). Elegant studies using complete ablation of hepatocytes have demonstrated the role of mTORC1 in cholangiocyte-to-hepatocyte transdifferentiation[52–54]. However, in the previous work, mTORC1 function was evaluated by inhibiting the pathway after liver injury. In contrast, we performed the inhibition in uninjured liver (Fig. 7G). Thus, our experimental setup uniquely demonstrates that mTORC1 signaling pre-injury can regulate cellular responses post-injury. It is intriguing to note that high mTORC1 signaling by itself does not lead to cholangiocyte transdifferentiation in the uninjured liver, but rather makes the cholangiocytes poised/plastic. We propose that mTORC1 signaling does not influence Notch levels under homeostatic conditions. This is supported by the fact that in a growing liver, mTORC1 activity is high (Fig. 7D–F), yet no spontaneous transdifferentiation occurs. However, upon injury, we hypothesize that elevated mTORC1 levels enable cholangiocytes to effectively downregulate Notch signaling, promoting transdifferentiation. This distinction is further supported when comparing 5 dpf and 10 dpf larval zebrafish. Previous studies, including our own, indicate that at 5 dpf, zebrafish are entering a fasting state[59–62], during which mTORC1 activity in cholangiocytes is low (Supplementary Fig. 12). At 5 dpf, it has been shown that PHx does not induce cholangiocyte-to-hepatocyte transdifferentiation, similar to adult zebrafish[16]. Notably, previous studies utilized *Tg(tp1:CreER)* for lineage tracing of cholangiocytes in 5 dpf larval and adult zebrafish, which may have influenced the observed absence of transdifferentiation. Nonetheless, these findings suggest that the mTORC1 pathway primes cholangiocytes to alter their identity in response to injury.

Further, mTORC1 activity is also strikingly higher in hepatocytes from 10 dpf larvae as compared to hepatocytes in adults (Figs. 7D, F) or 5 dpf larvae (Supplementary Fig. 12). Given that mTORC1 activation is highly responsive to protein uptake[63], mTORC1 signaling in hepatocytes may support protein metabolism at late larval stage, when the animals are being fed with rotifers that have high protein content[64]. It would be of interest to raise zebrafish on diets varying in protein and lipid content to evaluate mTORC1 levels and the cellular source of liver regeneration.

Potential upstream regulators of mTORC1 activity are growth factors and/or amino acids derived from exogenous food[63]. Notably, nutrition has been shown to regulate mTORC1 activity in the pancreatic ductal cells of zebrafish and regulate their transdifferentiation into endocrine cells[39]. Thus, nutrition, mTORC1, and ductal cell plasticity are intricately connected.

Similar to mTORC1, cholangiocytes at larval stage express *hhex*, *sox9b*, and *yap1* (Fig. 5F), which have been reported as regulators of the transdifferentiation process[48,50,51,65]. We, however, speculate that these

genes enhance the malleability of cholangiocytes, rather than directly pushing them towards a progenitor or hepatic fate. The exact nature of injury signal that induces the shift of cholangiocyte identity is missing from our work. Potential inductive signals include molecules released from dying or damaged hepatocytes, such as ATP or nitric oxide, or mechanical cues such as increase in blood flow[22]. In this regard, our models can be used to screen for signals that can induce cholangiocyte-to-hepatocyte transdifferentiation without injury in our setup. Thus, the study of liver regeneration in late larval stage zebrafish can inform us about the enhancers of cholangiocytes plasticity and modulators of cholangiocyte transdifferentiation, which would be critical information for developing interventions against liver damage and failure.

## Methods

### Zebrafish lines and husbandry

Wild-type and transgenic zebrafish from the outbred AB strain were utilized in all experiments. Zebrafish larvae were kept at 28.5 °C, with ~50 larvae per 30 ml of E3 medium (5 mM NaCl, 0.17 mM KCl, 0.33 mM CaCl2, 0.33 mM MgSO4, 1 mM HEPES) until 5 days post-fertilization (dpf). At 5 dpf, the larvae were fed daily with rotifers (100 μl/larva/day) and maintained at a density of around 30 larvae per 500 ml of 5 ppt fish water until 10 dpf. Subsequently, they were transferred to tanks with water flow and fed a dry fish flake diet twice daily. All zebrafish husbandry and experimental procedures for transgenic lines were conducted in accordance with institutional (Université Libre de Bruxelles) and national ethical and animal welfare guidelines and regulations, which were approved by the Ethical Committee for Animal Welfare (CEBEA) from the Université Libre de Bruxelles (protocols 864 N, 865 N, 877 N, 881 N, 882 N).

In this study, the following published transgenic lines were used: *Tg(Tp1bglob:H2B-mCherry)*[S939], also referred to as *Tg(EPV.Tp1-Mmu.Hbb:hist2h2l-mCherry)*[s939], abbreviated as *Tg(tp1:H2B-mCherry)*[35], *Tg(ubb:loxP-CFP-STOP-Terminator-loxP-H2B-mCherry)*[jh63 38] abbreviated as *Tg(ubi:CSHm)*[44], and *TgBAC(hand2:EGFP)*[pd24 66].

The following lines were newly generated by the I-SceI system: *Tg(fabp10a:creERT2; cryaa:CFP)*[ulb34], *Tg(fabp10a:H2B-mGreenLantern)*[ulb39] abbreviated as *Tg(fabp10a:H2B-mGL)* and *Tg(fabp10a:lox2272-loxp-nls-mTagBFP2-stop-lox2272-H2B-mGL-stop-loxp-mCherry-NTR; cryaa:mCherry)*[ulb33] abbreviated as *Tg(fabp10a:BBNTR)*.

### Generation of the Tg(fabp10a:creERT2), Tg(fabp10a:BB-NTR), and Tg(fabp10a:H2B-mGreenLantern) lines

To generate the fabp10a:lox2272-loxp-nls-mTagBFP2-stop-lox2272-H2B-mGL-stop-loxp-mCherry-NTR; cryaa:mCherry construct [abbreviated as fabp10a:BB-NTR], the dsDNA sequence of BB-NTR flanked with EcoRI/PacI was synthesized by GenScript Biotech and used to replace flag-SpiCee-mRFP1 in the fabp10a:flag-SpiCee-mRFP1;cryaa:mCherry construct[61] using restriction enzyme-based cloning with EcoRI/PacI following by ligation with T4 ligase.

The fabp10a:CreER[T2]; cryaa:mCerulean construct was generated by replacing *ins* promoter in the ins:creERT2; cryaa:mCerulean[67] with 2.1 kb fabp10a promoter obtained from fabp10a:Xpt-β-cat, cryaa:Venus vector (Addgene Plasmid #105127) using SacI/EcoRI.

To generate the fabp10a:H2B-mGreenLantern construct, we first obtained the fabp10a promoter from the fabp10a:CreER[T2]; cryaa:mCerulean vector by digesting it with EcoRI/PacI, and KpnI to remove the eye marker. Next, we amplified the H2B-mGreenLantern sequence from the fabp10a:BB-NTR vector using Terra™ PCR Direct Polymerase Mix (Takara, 639270) and primers flanked by EcoRI and PacI sites. Finally, we ligated this amplified sequence into the backbone. Primers used to amplify H2B-mGreenLantern sequence were the following:

TCGAGGCGCGCCGAATTCGCCACCATGATGCCAGAGCCAGCG AAGTCTGCTC (forward) and TCGAGGCGCGCCGAATTCGCCACCAT-GATGCCAGAGCCAGCGAAGTCTGCTC (reverse).

To generate transgenic lines, a solution containing the 20 ng/μl of the construct was mixed with I-SceI meganuclease enzyme and injected into one-cell stage embryos to facilitate transgenesis.

### Generation of TgKI(her2-p2a-EGFP-t2a-CreERT2)[KI150] and TgKI(her9-p2a-EGFP-t2a-CreERT2)[KI151] lines

Knock-in lines were generated following the protocol outlined in Mi et al.[44]. In brief, the following gRNA were used to target the genomic loci:

her2 (on reverse strand): 5′-TTAATTAAGAAAGGTCACCAGGG-3′
her9 (on reverse strand): 5′-TGAGATGCGCAAGTCTACCAGGG-3′

Chemically synthesized Alt-R-modified crRNA, tracrRNA, and nuclease-free duplex buffer were ordered from Integrated DNA Technologies (IDT). The crRNA: tracrRNA duplex solution was prepared by mixing 1 μl 10 μM crRNA stock solution, 1 μl 10 μM tracrRNA stock solution, and 8 μl nuclease-free duplex buffer and then incubating at 95 °C for 3 min in a thermocycler, followed by natural cooling at room temperature for 15 min. Afterwards, the Cas9/gRNA RNP ribonucleoprotein (RNP) solution was prepared by mixing 2 μl HiFi Cas9 protein (IDT) and 2 μl crRNA:tracrRNA duplex solution in 37 °C for 10 min.

To generate the donor dsDNA for knock-in, template sequence for p2a-EGFP-t2a-CreER[T2] was amplified using the following primer pairs:

her2 Forward Primer: TCAGAAATCGCCAAGCATGGATTGTGGA-GACCCTGGGGAAGCGGAGCTACTAACTTCAGC

her2 Reverse Primer: TTTATAAAAACAACACGTTTGGCTTTA ATTAAGAAAGGTCAAGCTGTGGCAGGGAAACCC

her9 Forward Primer: GGAGCAGAAAGCAATGAGCCGGTGTGGA-GACCCTGGGGAAGCGGAGCTACTAACTTCAGC

her9 Reverse Primer: ATGAAAACTTTATAAGTTCATATGAGATGC GCAAGTCTAAGCTGTGGCAGGGAAACCCTC

Primers with 5′ AmC6 modification were obtained from Integrated DNA Technologies and utilized to amplify the donor dsDNA.

Finally, the injection mix was prepared by mixing 2 μl Cas9/gRNA RNP, 5 μl donor dsDNA, and 0.8 μl phenol red (Sigma-Aldrich, P0290) and stored at 4 °C. The mix was prepared the day before injection.

The mix was injected into zebrafish embryos at the early one-cell stage. The overall mortality rate was around 50% and the dead embryos were sorted out in the following days. Mosaic F0 at 1 dpf were selected based on the green fluorescence under a wide-field fluorescence microscope LEICA M165 FC (Leica Microsystems). Positive mosaic F0 were put into the fish facility at 6 dpf. Adult F0 animals were outcrossed to WT animals to identify germline founder animals.

### 4-Hydroxytamoxifen (4-OHT) labeling

For consistent recombination efficiency across the experiments, an appropriate concentration of 4-OHT (MedChemExpress, HY-16950) was prepared from 10 mM stock in 100% DMSO and heated at 65 °C for 10 min to activate trans-4-OHT[68]. All 4-OHT treatments were carried out in the dark at 28.5 °C. The labeling and tracing experiments were performed under the following treatment conditions:

**CellCousin.** 10 μM 4-OHT treatment at 102 hpf ( ~ 4.5 dpf) for 24 h in 30 ml E3 medium with 30 larvae per petri dish.

her2 and her9 tracing: 20 μM 4-OHT treatment at 48 hpf for 30 h in 10 ml E3 medium in 6-wells plate with 10 larvae per well.

### Genetic partial ablation of hepatocytes

For partial ablation experiments, *Tg(fabp10:CellCousin)* zebrafish larvae were treated with freshly prepared 10 mM metronidazole (Mtz) (Thermo Scientific, #443-48-1) solution in 1% DMSO in E3 medium. The solution was vortexed continuously for 10 min to ensure complete dissolution of Mtz. Larvae were exposed to 10 mM Mtz at 10.5 and 11.5 dpf for two 16-h intervals with refreshment. Each treatment involved transferring 30 larvae to a 90 mm petri dish containing 30 ml of the

Mtz solution, protected from light. A 1% DMSO solution was used as the vehicle for all control animals. After each incubation, larvae were rinsed three times with E3 medium and then transferred to tanks with water flow, except between treatments when larvae were kept in 5 ppt fish water containing live prey (rotifers) to recover and feed. After the final washout, larvae were transferred to tanks with water flow and maintained on a dry food regime.

## Pharmacological treatments

Inhibition of Notch signaling pathway was performed using LY411575 (MedChemExpress, HY-50752). Working concentrations of LY411575 were prepared from a 50 mM stock solution dissolved in 100% DMSO. Zebrafish larvae were incubated with 50 μM LY411575 at 10.5 and 11.5 dpf for two 16-h intervals, or with 10 μM LY411575 at 10.5, 11.5 and 12.5 dpf for three 16-h intervals. Inhibition of mTORC1 signaling pathway was performed using Rapamycin (MedChemExpress, HY-10219). Working concentration (10 μM) of Rapamycin was prepared from 1 mM stock solution dissolved in 100% DMSO. Zebrafish larvae were incubated with 10 μM Rapamycin at 12.5 and 13.5 dpf for two 16-h intervals. For each incubation, the treatments were performed with refreshed solution protected from light. After each incubation, larvae were rinsed three times with E3 medium and then transferred to tanks with water flow, except between treatments when larvae were kept in 5 ppt fish water containing rotifers to recover and feed. After the final washout, larvae were transferred to tanks with water flow and maintained on a dry food regime, except after Rapamycin final washout larvae were kept in clean fish water tank until the surgery.

## Partial hepatectomy

For partial hepatectomy experiments, 10 days post-fertilization (dpf) late larval zebrafish were used. Fish were anesthetized in a 0.02% tricaine (MS-222) (Sigma-Aldrich, E10521) solution prepared in E3 medium within a petri dish. Once anesthetized, a small incision was made on the left ventral abdomen using Dumont #5 forceps (Fine Science Tools, 11295-10) to remove the skin and expose the liver. The left lobe was pinched in the middle, where it connects to the anterior loop, and carefully excised. Following resection, the fish were placed in a recovery tank with clean system water. Fish were monitored until they regained normal swimming behavior. For sham surgery, the incision on the skin was made without injuring the liver.

## Immunofluorescence

Zebrafish were euthanized by prolonged (25 minute) immersion in a 500 mg/l solution of tricaine (MS-222) (Sigma-Aldrich, E10521) and subsequently fixed in 2% paraformaldehyde (PFA) (Thermo Scientific, 28906) for at least for 24 h at 4 °C. After fixation, animals were washed five times for 5 min each in 0.3% PBSTx (0.3% Triton X-100 in PBS). For whole mount antibody staining, the skin on the lateral sides was carefully removed using fine forceps to expose the liver tissue enhancing permeabilization. Larvae at 10-17 dpf were permeabilized with 0.5% PBSTx for 30 min, while at 21 dpf they were permeabilized for 45 min. For antibody staining on cryosections, after fixation of euthanized larvae or adults in 2% PFA, livers were dissected and incubated in 30% sucrose in PBS for 24 h at 4 °C. Next, samples were embedded in cryomolds with OCT embedding matrix (Carl Roth, 6478.2) and frozen at −80 °C. Livers were cryosectioned into 12 μm thick sections and air-dried for 30 min. Sections were then washed three times for 5 min each in 0.3% PBSTx before blocking. Blocking was performed in 0.1% PBSTx with 4% goat serum and 1% bovine serum albumin (BSA) (VWR, 422361 V). Primary antibody incubation was performed for 48 h at 4 °C using the following primary antibodies: mouse anti-Anxa4/2F11 (1:100, Abcam, ab71286), rabbit anti-mCherry (1:1000, CST, 43590) and for 3 h at 37 °C for mouse anti-pS6 (1:200, CST, 2215S). Secondary antibodies (Alexa Fluor 647 donkey anti-mouse, Cy3 donkey anti-rabbit) were added at 1:1000 to PBSTx 0.1%

with 4% serum (goat) and 1% BSA and incubated for overnight at 4 °C or 1 h at 37 °C. DAPI was included in the secondary antibody solution. After staining, livers from whole-mount immunostained larvae were dissected under stereo microscope. Sections and whole-mount immunostained livers were mounted in Fluoromount-G™ mounting medium (Invitrogen, 00-4958-02) prior to imaging.

## Imaging

Before in vivo confocal imaging, larvae were anesthetized with 0.02% tricaine (MS-222) (Sigma-Aldrich, E10521) for 1 min and then mounted in 1% Low-Melt Agarose (Lonza, 50080) containing 0.02% tricaine. Imaging was performed on a glass-bottomed dish FluoroDish™ (WPI, FD3510-100) using Zeiss LSM 780 confocal microscope platform. Livers were imaged using a 40x/1.1 water immersion lens for larvae up to 7 dpf and a 25x/0.8 immersion correction lens for larvae older than 9 dpf. The imaging frame was set to 1024 × 1024 pixels, with a distance of 6 μm between confocal planes for Z-stack acquisition. Samples were excited at 405 nm for mTagBFP2, 488 nm for mGreenLantern, 543 nm for mCherry, and 633 nm for Alexa Fluor 647 with fluorescence collected in the respective ranges of 426-479 nm, 497-532 nm, 550-633 nm, and 641-735 nm. Whole-mount livers were imaged in stereo microscope Leica M165 FC with DFC 7000 T camera platform using a 2.5x lens and filter set ET GFP. Imaging was recorded in Leica Application Suite X (v1.9). For confocal imaging of whole-mount livers at 21 dpf (11 dpi), individual livers were mounted in Fluoromount-G™ mounting medium (Invitrogen, 00-4958-02) on glass slides (Menzel Gläser, SuperFrost) (VWR, 631-0705) using #1 cover slip (Carl Roth, H878.2). Images were taken on Zeiss LSM 780 confocal microscope using 10x/0.45 air lens. The imaging frame was set to 1024 × 1024 pixels, with a distance of 6 μm between confocal planes for Z-stack acquisition. Samples were excited at 488 nm for mGreenLantern.

## EdU incorporation assay

EdU staining was performed using the Click-iT Plus EdU Cell Proliferation Kit for Imaging, Alexa Fluor™ 647 dye (Thermo Scientific, C10640) as described in Madakashira et al.[69] with minor modifications. A 200 mM stock solution (1:20) of EdU was prepared by dissolving 100 mg of EdU (Jena Bioscience, CLK-N001-100) in 2 ml DMSO and distributed in 50 μl aliquots. The aliquots were stored in −20 °C. A 10 mM EdU with 5% DMSO solution was prepared in E3 to reach a total volume of 250 μl for treating 5-7 larvae. Larvae were placed in a 50 ml conical tube containing 250 μl of the EdU solution, which was then incubated on ice for 4 min. Immediately after the incubation, 50 mL of pre-warmed (28 °C) E3 media was added, and the tube was laid on its side for 5 min on a shaker to allow larvae recovery. Once recovered, the falcon tube was transferred to a 28 °C incubator (kept on its side) for a 4-h chase period. Following the chase, larvae were fixed by 4% PFA with 1% Triton X-100 for 90 min at room temperature. The samples were then washed twice with PBS and dissected under a microscope to remove the tail, skin, eyes, and brain. The remaining tissue was transferred to a 2 ml tube containing PBS, which was then replaced with PBS with 0.5% Triton-X for 10 min. During this period, EdU detection mix was prepared.

For EdU detection, the EdU detection mix was prepared by mixing 1X Click-iT reaction buffer, Copper protectant and Alexa Fluor picolylazide according to the manufacturer's protocol. No EdU additive was added. This gave a 450 μl solution per sample that was protected from light. Tissue samples were transferred to 0.5 ml tube and quickly rinsed twice in 3% BSA in PBS. Then, samples were incubated in 225 ul of prepared EdU detection mix for 5 min on a shaker, protected from light. During this time, a 1X Click-iT EdU buffer additive solution was prepared fresh (50 μl per sample). To each sample, 25 μl of 1X additive solution was added and the samples were future incubated for 20 min on shaker protected from light. Following this, the EdU detection solution was removed. Then, remaining 1X additive (25 ul) and

remaining EdU detection mix (225 ul) was mixed and 250 ul of solution was immediately added for an additional 25 min of shaking, protected from light. Samples were then washed 1-2 times with PBS, transferred to a 2 ml tube with fresh PBS (samples can be stored at 4 °C overnight if needed). For optional nuclei counterstaining, samples were incubated in 0.5% PBST with 1:1000 DAPI or Hoechst for 60 min at room temperature or overnight at 4 °C. Finally, the liver was dissected in PBS and mounted onto a slide for imaging.

## Image analysis

Image analysis was performed using Imaris 10.0 (Oxford Instruments). The Imaris software was utilized to calculate liver volume by manually selecting liver segments using the 'surface' module. For further analysis, the channels present in the images were masked by the liver volume surface to avoid auto-fluorescence detection in the quantifications.

Quantifications related to the number of hepatocytes in the *Cell-Cousin* model were performed by manually thresholding the relevant fluorescent signal (nuclear or cytoplasmic) and separating objects based on intensity, with an estimated diameter set to 6 μm. To detect double-positive cells in the *CellCousin* model, objects were filtered and classified based on the mean relative fluorescent signal in the channels.

For quantifications based on nuclear fluorescent signals, excluding the CellCousin model, the 'spots' module was used. The expected diameter was set to 6 μm for hepatocytes and 5 μm for cholangiocytes. Colocalization analysis was performed based on the spot-to-spot shortest distance distribution of less than 2 μm.

Quantifications related to the length ratio of lobes, liver volume and pS6 intensity were performed using Fiji (Image J), version 2.16.0[70].

## Western blotting

RIPA buffer (CST, #9806) was used to extract proteins from 10 days post-fertilization zebrafish larvae (n = 10 larvae) and adult zebrafish liver from male and females (n = 2). To protect the proteins, Halt Protease and Phosphatase Inhibitor Cocktail (Thermo Fisher Scientific, #78440) was added to the RIPA buffer. The proteins were quantified using BCA protein assay kit (Thermo Fisher Scientific, #23225). After separation with polyacrylamide gel, 20 or 25 ug of protein were transferred to a 0.2 um nitrocellulose membrane (BioRad, #1620112). Primary antibodies (anti-pS6 (1:1000, CST, 2215S) and GAPDH (1:50000, ProteinTech, 60004-1-Ig) were diluted in 5% milk-blocking buffer, and for the detection of proteins goat anti-rabbit IgG (Dako Agilent, #P044801-2) or goat anti-mouse IgG (Dako Agilent, #R048001-2) were used. The detection of immunoreactive bands was performed using a Western blot imaging system (Amersham Image-Quant 800, Cytiva Life Science).

## Preparation of cell suspension for sorting and single-cell RNA sequencing

Zebrafish were euthanized in 500 mg/l solution of tricaine (MS-222) (Sigma-Aldrich, E10521). The entire liver was dissected out of the body using Dumont #5 forceps (Fine Science Tools, 11295-10). Single-cell suspension was prepared by adapting the cell dissociation protocol outlined in Singh et al.[71]. All steps were performed with tubes and pipette tips coated with 1% BSA in PBS prior to the sample preparation[72]. Briefly, the liver was dissociated into single cells by incubation it in TrypLE (Thermo Fisher, 12563029) at 37 °C in a benchtop shaker set at 1000 rpm for 15 min. Following dissociation, TrypLE was inactivated with goat serum. To remove undissociated chunks and debris, the solution was passed through a 40 μm cell strainer (Miltenyi Biotec, 130-041-407). Cells were pelleted by centrifugation at 500 g for 5 min at 4 °C with soft-stop setting. The supernatant was carefully discarded and the pellet re-suspended in 500 μl of PBS. To remove dead cells, Calcein violet (Thermo Fisher, C34858) or Draq7 (Thermo Fisher, D15105) was added at a final concentration of 1 μM and 3 μM, respectively, and the cell suspension was incubated at room temperature for 20 min. The single-cell preparation was sorted with FACS-Aria II (BD Bioscience) using appropriate gates, including excitation with UV (405 nm) or a 633 nm laser for identifying live cells with Calcein+ or Draq7-, respectively. FACS was performed using a 100 μm nozzle, and a minimum of ~15,000 live cells per condition were collected into BSA-coated tubes containing 1% BSA in PBS. Sorting time did not exceed 15 min. FACS data was analysed using FCSalyzer, Version 0.9.22-alpha (https://sourceforge.net/projects/fcsalyzer/).

## Single-cell transcriptome profiling

For single-cell RNA-sequencing using the 10x Genomics platform, the cell suspension was adjusted with PBS to a density of 300 cells/μl and diluted with nuclease-free water according to the manufacturer's instructions to yield 7,000 cells. Subsequently, the cells were carefully mixed with reverse transcription mix before loading the cells on the 10x Genomics Chromium system (Single Cell 3' v3). After the gel emulsion bead suspension underwent the reverse transcription reaction, emulsion was broken and DNA purified using Silane beads. The complementary DNA was amplified with 12 cycles, following the guidelines of the 10x Genomics user manual. The 10x Genomics single-cell RNA-seq library preparation—involving fragmentation, dA tailing, adapter ligation, and indexing PCR—was performed based on the manufacturer's protocol. After quantification, the libraries were sequenced on an Illumina NextSeq 550 machine. A custom Cell Ranger index was generated from the Ensembl GRCz11 genome sequence and annotation filtered with 'mkgtf' command of Cell Ranger (options: '−attribute = gene_biotype:lincRNA −attribute = gene_biotype:antisense), as well as the pseudo mTagBFP2, H2B-mGL, mCherry-NTR sequences in these cells using the mkref command. This index was used to map reads and generate gene expression matrices using the 'count' command of the Cell Ranger software (v.7.1.0) provided by 10x Genomics with the option "−include-introns" set to False (all other options were used as per default).

## Analysis of single-cell RNA sequencing data

The raw data generated from 10x Chromium pipeline was processed using Scanpy (v1.10.2)[73] according to the recommended analysis workflow. Initially, the raw data were loaded, and doublets were predicted separately for each sample using *scanpy.external.pp.scrublet()* function from Scrublet[74]. The resulting objects were concatenated using the *AnnData.concatenate()* function, resulting in a total 49,199 cells. Low quality cells were filtered out based on following criteria: cells with less than 500 or more than 8000 genes, and cells with more than 15% mitochondrial gene content. Additionally, genes detected in less than 3 cells were excluded. The filtered data were then log-normalized and scaled. Highly variable genes were selected with *scanpy.pp.highly_variable_genes()* using top 2000 genes and principal component analysis (PCA) was performed with the selected genes. The top 50 principal components (PCs) were selected and used to compute a neighborhood graph. UMAP was computed using default parameters. Unsupervised clustering was computed using Leiden algorithm with 'flavor=igraph' and 'n_iterations=2'. Clusters having less than 0.09 doublet score in addition to the cells not predicted as 'doublets' were used for downstream analysis (30,856 cells and 22,976 genes).

Batch correction was performed using *scanpy.external.pp.harmony_integrate()* function from Harmony[75], and the final clustering was computed using top 20 PCs with 15 neighbors. A resolution of 1.2 was used for Leiden clustering. Marker genes for each cluster were identified and used to classify cell types. To label mTagBFP2+ cells, a threshold of 0.1 was set on normalized expression of mTagBFP2, and cells were annotated with their respective condition information as mTagBFP2+ cells or spared hepatocytes (mTagBFP2-).

Pearson correlation and hierarchical clustering was computed using *scanpy.tl.dendogram()* on the results of PCA components and visualized with *scanpy.pl.correlation_matrix()* between mTagBFP2+ cells, spared hepatocytes, control hepatocytes and cholangiocytes. Additionally, a Pearson correlation matrix was computed on the top 2000 highly variable genes based on average expression per cluster using *corr()* function to confirm the results of the PCA-based correlation matrix.

For comparative analysis, we compared our control-uninjured larval zebrafish dataset to a public adult zebrafish liver dataset (ID: GSE 181987; Samples: GSM5515731, GSM5515732, GSM5515733)[47]. The two raw datasets were concatenated, filtered and annotations were transferred from Morrison et al. resulting in a combined dataset of 23,014 cells and 19,376 genes. After log-normalization, highly variable genes were identified, PCA and batch correction were performed using the first 20 PCs with 15 neighbors, and cluster analysis was finalized. For downstream analysis, cholangiocytes from larval and adult stage were retained, and differential gene expression analysis was performed using the Wilcoxon test.

### Subclustering, pseudotime trajectory analysis and differential gene expression analysis

For subclustering of larval data, cholangiocytes from uninjured (control) samples were subsetted from the dataset and processed following standard normalization procedures. Dimensionality reduction was performed by selecting the top 2000 highly variable genes, followed by principal component analysis using the top 22 principal components. A neighborhood graph was computed with 'n_neighbors=15', and clustering was performed at a resolution of 0.4. Clusters enriched in acinar cell-specific markers (indicative of contamination) and those exhibiting high ribosomal and heat-shock gene expression (potentially damaged cells) were excluded from further analysis.

To reconstruct the differentiation trajectory, we used Monocle3[76] on the combined dataset of injured and uninjured livers, as pseudo-temporal inference requires a continuum of cellular states that are not all present in control samples alone. Specifically, transitional or intermediate states appear only upon injury and were essential for trajectory learning. Clustering was performed using the *cluster_cells()* function with a resolution parameter of 1e-4, and trajectory graph learning was conducted using the *learn_graph()* function with 'minimal_branch_len=10'. Cells were then ordered along the inferred trajectory using the *order_cells()* function with default parameters. The root node for pseudotime inference was manually defined within the IHD/intermediate cholangiocyte cluster from control samples to ensure biologically relevant reconstruction of the transdifferentiation process.

For visualization of subclusters within the cholangiocyte population, UMAP coordinates computed in Scanpy were used in place of the default Monocle3 layout to maintain consistency across all analyses. The "cholangiocytes rest" cluster refers to the population of cholangiocytes derived from injured samples.

For subclustering of adult data, cholangiocytes from uninjured (control) samples were subsetted from the Morrison et al. dataset[47]. This represents the zChol1, zChol2 and zAgr2+ clusters from the dataset. After log-normalization and identification of highly variable genes, dimensionality reduction was performed using the top 20 principal components. A neighborhood graph was computed with n_neighbors=15, and Leiden clustering was applied at a resolution of 0.4. Subclusters were annotated based on expression of known marker genes, including *her2* for IHD/intermediate cholangiocytes and *clndh* for luminal cholangiocytes.

To enable direct comparison with the larval dataset, the annotated IHD/intermediate and luminal cholangiocyte subclusters from both larval and adult livers were extracted and merged. Differential gene expression analysis was performed between larval and adult subclusters using the Wilcoxon rank-sum test as implemented in Scanpy. Significantly upregulated genes were identified using a false discovery rate (FDR) threshold of 0.05.

### Statistical analysis

Statistical analysis was performed using GraphPad Prism software (version 10.2 GraphPad Software, San Diego, CA). When comparing multiple groups, we used one-way ANOVA test followed by Kruskal-Wallis multiple comparison test. When comparing two groups, we used a Two-tailed Mann-Whitney test. No data was excluded from analysis. Blinding was not performed during analysis.

### Reporting summary

Further information on research design is available in the Nature Portfolio Reporting Summary linked to this article.

## Data availability

The raw files and raw count table from deep sequencing can be accessed at Gene Expression Omnibus (GEO) with accession number GSE272484. We provide the annotated single-cell RNA-seq datasets via an interactive application, built with the iSEE package[77] and available at http://shiny.imbei.uni-mainz.de:3838/iSEE_CellCousin. Plasmids generated in this manuscript have been deposited to Addgene: fabp10a:lox2272-loxp-nls-mTagBFP2-stop-lox2272-H2B-mGL-stop-loxp-mCherry-NTR; cryaa:mCherry: Plasmid #230043; fabp10a:CreERT2; cryaa:mCerulean: Plasmid #230044; fabp10a:H2B-mGreenLantern: Plasmid #230045. Raw images and image analysis files are available upon request to the corresponding author. Source data are provided with this paper.

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

## Acknowledgements

We thank Dr Kirsten Sadler Edepli (NYU Abu Dhabi, UAE), Dr Donghun Shin (University of Pittsburgh, USA) and Dr Nikolay Ninov (Center for Regenerative Therapies, Dresden (CRTD), Germany) for comments on the manuscript. We thank Dr Elena Magnani (NYU Abu Dhabi, UAE) for advice on EdU staining. We thank the members of IRIBHM Fish Facility, Christine Dubois from FACS facility, and M Martens and JM Vanderwinden from the Light Microscopy Facility for technical assistance at ULB. This work was supported by the Deutsche Forschungsgemeinschaft (DFG, German Research Foundation) Projektnummer 318346496 - SFB1292/2 TP19N to FM. ENG is a Research Associate of the Fonds de la Recherche Scientifique (FNRS), Belgium. The work was supported by FNRS grants 40006730 (ASP) to SEE, 40021615 (FRIA) to GGH, and 40005588 (MISU-PROL), 40013427 (CDR), 40027730 (CDR) and 40020360 (PDR) to SPS, and funding from Université libre de Bruxelles, Jaumotte-Demoulin Foundation, and Ramalingaswami Re-entry Fellowship from Department of Biotechnology (DBT), India to SPS.

## Author contributions

S.E.E.: investigation, visualization, methodology, writing—original draft, review, and editing. J.M.: methodology, resources, and writing—review and editing. M.P.M.: investigation and resources. CP, FM, ENG: resources. RM: methodology and writing—review and editing. G.G.H., I.E.Z., D.B., A.L., F.L.: investigation. O.A.: supervision, methodology, and writing—review and editing. S.P.S.: conceptualization, supervision, funding acquisition, project administration, and writing—original draft, review, and editing.

## Competing interests

The authors declare no competing interests.
