## [Transparent Peer Review file · Nature Communications]

Cholangiocytes contribute to hepatocyte regeneration after partial liver injury during growth spurt in zebrafish

Corresponding Author: Professor Sumeet Singh

Version 0:

Reviewer comments:

Reviewer #1

(Remarks to the Author)

In this manuscript, Eski and co-authors designed two partial liver injury models in zebrafish and found that cholangiocyte-to-hepatocyte transdifferentiation is the primary mechanism of liver regeneration during rapid growth stages. They also found that mTORC1 regulates the plasticity of cholangiocytes. However, these findings are similar to a recently published paper, in which restricted hepatocyte ablation is sufficient to trigger local BEC aggregation and transdifferentiation (Ambrosio et al., *Development* 2024; doi:10.1242/dev.202217). Up to now, it is well known that cholangiocytes could transdifferentiate into hepatocytes in several liver injury models, and the roles of mTORC1 in cholangiocyte-to-hepatocyte transdifferentiation have been characterized in zebrafish and mice. So, the observation in this study that partial liver injuries, including partial hepatectomy and hepatocyte ablation, stimulate transdifferentiation of cholangiocytes has been identified and characterized, thus lacking novelty.

Major points

1. The manuscript describes cholangiocyte-to-hepatocyte transdifferentiation as the primary mechanism of liver regeneration during rapid growth stages. However, only a partial number of newly regenerated hepatocytes come from her2+ cells at 8 dpi after PHx in Fig.4F and G. According to the data, the H2B-mGL+ hepatocytes should be the primary cell sources for liver regeneration (Fig. 4D and F).
2. Fig.1E and G show that nearly all the newly regenerated hepatocytes (mTagBFP2+) were derived from non-hepatocytes. Meanwhile, less than half of hepatocytes marked by mCherry-NTR after 4-OHT treatment were ablated with Mtz incubation (Fig. 1F). What about the other half of hepatocytes during liver regeneration? Did the mGL+ hepatocyte proliferate, migrate or die during liver regeneration?
3. Most de novo regenerated hepatocytes are nls-mTagBFP2+ in Fig. 2F (non-hepatocyte-derived), but only partial newly regenerated hepatocytes are derived from her2+ cholangiocytes (Fig.4D-G). These data indicate other non-hepatocyte cell types besides cholangiocytes are also involved in hepatocyte regeneration. The authors need to address this important point.
4. The manuscript showed that the inhibition of Notch signalling led to the spontaneous emergence of hepatocytes. At the same time, the activities of mTORC1 in the larval cholangiocytes was stronger than those in adults. What is the relationship between Notch and mTORC1 in regulating cholangiocyte plasticity?

Minor points

1. It is essential to assess the ratios of non-hepatocyte-derived hepatocytes among all hepatocytes during liver regeneration.
2. Western blot is necessary to demonstrate the expression levels of pS6 in larval and adult zebrafish liver, as shown in Fig. 5D.
3. Could the inhibition of Notch signalling promote the conversion of cholangiocytes into hepatocytes in adult liver after PHx?

Reviewer #2

(Remarks to the Author)

The work by Eski investigates the regenerative response of the liver during a phase of rapid growth, as seen in young animals. Using zebrafish as a model, they employ two different ablation strategies, ablation of about 50% hepatocytes throughout the liver and PHx by surgical lobe resection. Based on genetic lineage tracing and slow histone turn-over tracing, they propose that new hepatocytes originate from cholangiocytes, rather than remaining hepatocytes or a mixed origin, as seen in adults or early larval zebrafish. Single cell transcriptomics are employed to corroborate cholangiocytes trans differentiation as source. Finally, these data further implicate enhanced mTORC1 signaling as mediator of this cholangiocyte plasticity during heightened growth phases, supported by subsequent inhibitor treatments.

Overall, the manuscript is well written and supported by imaging of excellent quality and appropriate quantifications. They also developed interesting new methodologies, including transgenic animals. The findings are overall novel and exciting, since little is known about mechanisms of postnatal/-embryonic regeneration. Therefore adding valuable insights to the hepatology field, and generally very interesting to the regeneration

1. De novo hepatocyte formation after partial hepatocyte ablation exclusively of cholangiocyte origin is an exciting proposal, and contrasts most liver regeneration in mammalian and zebrafish models, in which healthy hepatocytes contribute a substantial fraction of new cells by proliferation. To substantiate this intriguing interpretation and this key point of this work, complementary proliferation studies focusing on the different stages and cell types need to be performed, because current lineage tracing does for instance not reveal hepatocyte proliferation.

2. Here, it is notable that recombination is already initiated 5-6days before ablation, so that relatively large clones of NTR-mcherry form (see eg Fig 2A).

What is the recombination pattern and how does regeneration look like when recombination is induced shortly before NTR-ablation or PHx?

3. After PHx, which parts of the liver produce new hepatocytes? Surprisingly, this is not clearly described or shown. This is generally an important point, and specifically because after PHx in adult zebrafish new cells come either from a blastema or from compensatory growth and the reasons are not entirely clear yet (Oderberg and Goesling, 2021).

Schematic (Fig2D) indicates that the surgically removed part is replaced by epimorphic regeneration, in the sense that the missing tissue regrowth is exclusively formed by TagBFP2 cells. Is this what is observed? The corresponding experimental results (Fig2F) are difficult to interpret, since it is unclear where in the image the resection site is located and which part of the liver is exactly shown.

It should be clearly addressed where in the liver, related to the resection site new hepatocytes from cholangiocyte transdifferentiation form. Analyzing and quantifying different regions in the same liver, e.g. left, right lobe and connecting part. Does the right lobe show a compensatory response contributing new hepatocytes to the regeneration of the liver? Like for the chemical NTR-model above, complementary proliferation studies are necessary to pinpoint the contributing of regional sources and cell types. Would that be expected given the 'primed' status of the cholangiocytes?

4. Further to the source of new cells following PHx, where is the resection site in Figure 4D and F? Where are the groups of mCherry cells located in the provided images in relation to it? Also please explain why there seem to be small focal clones of hepatocytes derived from cholangiocytes, rather than a more wider response throughout the remaining organ as one might expect when larval cholangiocytes show a generally higher plasticity? What is the spatial distribution of such clones?

5. PHx, the impaired growth response of the third lobe is interesting, can the authors exclude that they do not remove some of the ventral lobe progenitors? Similar right lobe PHx should be performed as appropriate control for this.

6. Also, please elaborate on the proposed morphological response? Is it possible that the anlage for the ventral lobe gets damaged during the left lobe surgery – representing the reason for smaller lobe?

Length measurement, even if only an approximation of regeneration, suggests that the left lobe recovers by an epimorphic mechanism.

7. The authors make a strong point about unlimited food availability during the growth phase, as well as increased expression of components of mTORC1 signaling. Is this a real link? It should be tested whether limited feeding has an effect on the expression of mTORC1 signaling and the regeneration response.

Since in the late embryo/early larvae hepatocytes also contribute to regeneration, what makes their response different? Finally, hepatocytes and cholangiocytes both express mTORC1 targets, what is the role in hepatocytes in this context? Do the inhibitor treatments give an indication? The authors should at least discuss possible mechanisms for different functions.

8. Given the use of the her2-driver line in this study, validation, including quantification, of cell type specificity should be provided (similar to figS8).

The same should be done for the her9-driver line. The images are excellent, though unclear which other cell types are marked and at which frequency.

9. Text and Figure S8: states that the her9+ lineage mainly contains cholangiocytes. Such a statement should be supported by appropriate quantification, which is essential for the conclusions that use this line.

Minor points:

1. Fig4D,F – please show consistently the larger magnifications
2. For the cholangiocyte response, the authors use several times the term reprogramming, which is generally associated with the removal or remodeling of epigenetic marks, which they do not address. Hence consistent use of transdifferentiation seems more appropriate.
3. Please indicate what the data in Figure 1G, 2G and S7 E,G are normalized to.
4. Figure 4: mention of non-existing panel H in Figure legends.
5. Fig4D,F – indicate the resection site.
6. For a number of imaging data, it would be very helpful to see magnifications of the data, e.g. fig 2F or 5G,

Version 1:

Reviewer comments:

Reviewer #1

(Remarks to the Author)

Again, I would repeat the key point of my original comments: The cholangiocyte-to-hepatocyte transdifferentiation has been intensively studied so that this manuscript is devoid of novelty.

Besides, the fact that this transdifferentiation only occurs in larvae, but not in adult zebrafish and mice after PHx, diminishes the significance of this finding.

Furthermore, some key tools used in this study are not optimal. For example, the her9 and her2 transgenes, because they label not only cholangiocytes but also other cell types such as satellite cells.

Therefore, I would not support publication of this paper at Nature Communications.

Reviewer #3

(Remarks to the Author)

The authors addressed several prior reviewers' comments and made additional important observations. These new data strengthen the paper and enhance its novelty. However, I have additional questions and comments:

1. Cholangiocyte Contribution to Hepatocyte Regeneration

The main conclusion of the paper is that cholangiocytes contribute to hepatocyte regeneration during growth spurts in zebrafish, even when a significant number of hepatocytes remain after injury. Cholangiocytes are heterogeneous based on their localization, morphology, and gene signatures. Which cholangiocyte population(s) specifically contribute to hepatocyte regeneration in the two injury models described in the paper? Furthermore, the current data do not provide sufficient evidence to exclude the possibility that hepatic progenitor cells, rather than differentiated cholangiocytes, contribute to this regeneration. Additional experiments should be conducted. Otherwise, the main conclusion from this paper is too strong of a statement.

2. Comparative Analysis of Partial Hepatectomy (PHx)

The authors show that partial hepatectomy (PHx) induces cholangiocyte transdifferentiation at 10 days post-fertilization (dpf), whereas previous studies reported that PHx at 5 dpf or in adult zebrafish does not induce cholangiocyte transdifferentiation. It is important to point out that different transgenic zebrafish models were used in these studies, which may explain the differing outcomes.

3. Control for PHx Experiments

What control was used in the PHx experiments? A sham control should be included to ensure that the surgical incision and removal of the skin do not influence ventral lobe growth independently of the hepatectomy.

4. Measurement of Liver Size

The exclusion of compensatory growth after PHx is primarily based on lobe length measurements (Figure 4D–H). However, liver area or volume measurements would be more appropriate to assess liver size accurately.

5. Identification of Her9-Lineage Cells

The authors concluded that 30% of her9-lineage cells are hepatic stellate cells. However, in Figure S8E, it is evident that all hand2:GFP+/Her9+ double-positive cells are located at the liver surface within the mesothelium. These cells are not hepatic stellate cells. The authors should revise this conclusion accordingly.

Additional comments regarding critique 1 (Cholangiocyte Contribution to Hepatocyte Regeneration):

To date, all published work on zebrafish liver regeneration involving cholangiocyte-to-hepatocyte transdifferentiation relies on Tp1-Cre or Tp1-CreERT2 to lineage trace existing cholangiocytes. However, Tp1, a Notch-responsive element, is not a cholangiocyte-specific promoter. It can also label hepatic progenitor cells. Additionally, the cholangiocyte markers commonly

used in zebrafish studies, such as 2F11, *alcam*, and *epcam*, are not exclusively expressed in differentiated cholangiocytes. This represents a significant limitation of these studies, which should be properly addressed.

Determining whether hepatic progenitors or differentiated cholangiocytes give rise to regenerating hepatocytes would require the development of additional tools and may be beyond the scope of this paper. However, I believe the single-cell RNA-seq data presented by the authors could provide valuable insights into this question. Within the identified “cholangiocyte” population, are there distinct subgroups? Specifically, are there more differentiated cholangiocyte subpopulations that express cholangiocyte-specific transporters and metabolic genes? Conversely, are there “cholangiocytes” that appear less mature and exhibit more progenitor-like properties?

Acknowledging the heterogeneity within Tp1+ populations and recognizing that different subsets may contribute differently to hepatocyte regeneration is a critical first step toward understanding the key cellular players in this process. Thus adding additional analysis of the scRNA-seq data will strengthen the paper.

Version 2:

Reviewer comments:

Reviewer #3

(Remarks to the Author)

The author's responses to reviewers' critiques have raised more questions as detailed below.

1. Newly generated data on liver regeneration after partial hepatectomy in juvenile mice.

The observation in Figure R2 neither supports nor refutes the authors' conclusion that cholangiocytes contribute to hepatocyte regeneration in juvenile mice following PHx. The percentage of CK19;HNF4 α double-positive hepatocytes is not reported, nor is the number of animals analyzed. Moreover, in the absence of lineage tracing, the presence of these double-positive cells could also suggest that hepatocytes contribute to cholangiocyte regeneration after PHx.

2. The authors presented new scRNA-seq data demonstrating four cholangiocyte clusters in the control zebrafish liver: one enriched for *krt4* (representing luminal ductal cells), two clusters comprising a mix of intrahepatic duct (IHD) and intermediate ductal cells, and a proliferative cholangiocyte cluster. Using pseudotemporal analysis, they proposed that IHD and intermediate ductal cells are the primary source of transdifferentiation.

However, the data in Supplemental Figure 6 raises more questions. In Fig. 6F, the “cholangiocytes rest” cluster (grey) appears to be more poised to transdifferentiate to hepatocytes than the other four cholangiocyte clusters, yet this cluster is not discussed in the text. The authors also refer to their previous scRNA-seq study of the adult zebrafish liver, in which IHD and intermediate ductal cells were also identified. Notably, PHx at the adult stage does not lead to cholangiocyte-to-hepatocyte transdifferentiation. Comparing the transcriptomes of IHD and intermediate ductal cells in juvenile versus adult livers would be crucial for uncovering the distinct mechanisms underlying liver regeneration at these two stages.

*Cholangiocytes contribute to hepatocyte regeneration after partial liver*
*injury during growth spurt in zebrafish.*

**Sema Elif Eski, Jiarui Mi, Macarena Pozo-Morales, Gabriel Garnik Hovhannisyan,**
**Camille Perazzolo, Rita Manco, Imane Ez-Zammoury, Dev Barbhaya, Anne Lefort,**
**Frédéric Libert, Federico Marini, Esteban N. Gurzov, Olov Andersson, Sumeet Pal**
**Singh***

* Corresponding author: sumeet.pal.singh@ulb.be

We thank the reviewers for the encouraging comments. We have attempted to address
all the issues outlined by the reviewers. We thank the reviewers for raising the concerns,
which were all valid. Addressing the comments has, we think, strengthened the
manuscript, and for this we appreciate the constructive feedback received from the
reviewers.

Particularly, in the two main areas, the reviewer's feedback has been greatly beneficial.
Firstly, in response to the comments, we have carried out a detailed, quantitative
analysis of cellular proliferation in response to partial ablation and partial hepatectomy
(PHx). In response to partial ablation, we observe a strong induction of cell-cycle in
cholangiocytes. However, we do not observe any increase in proliferation in the spared
hepatocytes, underscoring the specific response of cholangiocytes to the injury. In
response to PHx, we observe an increase in hepatocyte proliferation. This, however, is

much smaller and shorter than the enhancement in cholangiocyte proliferation. Thus, in
both injury models, cholangiocytes display a robust increase in cell-cycle.

Secondly, in response to the reviewer's comments, we have improved the presentation
of our images. We have exchanged full liver images with zoom-in images, and we have
opted to showcase a single confocal plane rather than maximum intensity projection
images. This significantly increased the clarity of image presentation.

We thank the reviewers for their insightful feedback that helped improve the manuscript
scientifically and visually.

For the revision, our response is in blue. To highlight specific image panels from the
manuscript, we have reproduced them in the response letter. In this case, they are
labeled with the numbers from the manuscript. Images specific to the rebuttal letter are
labeled as Figure Rn.

**REVIEWER COMMENTS**

**Reviewer #1 (Remarks to the Author):**

In this manuscript, Eski and co-authors designed two partial liver injury models in zebrafish and
found that cholangiocyte-to-hepatocyte transdifferentiation is the primary mechanism of liver
regeneration during rapid growth stages. They also found that mTORC1 regulates the plasticity
of cholangiocytes. However, these findings are similar to a recently published paper, in which
restricted hepatocyte ablation is sufficient to trigger local BEC aggregation and
transdifferentiation (Ambrosio et al., Development 2024; doi:10.1242/dev.202217). Up to now, it
is well known that cholangiocytes could transdifferentiate into hepatocytes in several liver injury
models, and the roles of mTORC1 in cholangiocyte-to-hepatocyte transdifferentiation have been
characterized in zebrafish and mice. So, the observation in this study that partial liver injuries,
including partial hepatectomy and hepatocyte ablation, stimulate transdifferentiation of
cholangiocytes has been identified and characterized, thus lacking novelty.

We would like to thank Reviewer #1 for their thorough evaluation and constructive
comments on our manuscript. We appreciate the acknowledgment of our work in
designing new liver injury models in zebrafish to explore the mechanisms of liver
regeneration.

In response to the points regarding novelty, we cite Ambrosio et al. in the original
manuscript and agree that their findings align well with our partial ablation model.

However, our work expands on these results in significant ways: 1. we show
transcriptional convergence between *de novo* hepatocytes and spared hepatocytes
using single-cell RNA sequencing, and 2. we employ lineage tracing with the Cre/Lox
system to confirm cholangiocyte-to-hepatocyte transdifferentiation. Further, we develop
and employ a partial hepatectomy (PHx) model to demonstrate cholangiocyte-to-
hepatocyte conversion in an injury model where all cell populations of the liver are
impacted. In contrast, Ambrosio et al. hypothesize that transdifferentiation is induced
specifically by hepatocyte loss. Finally, the conversion of cholangiocytes into
hepatocytes upon PHx is absent in mice (1), and, to our knowledge, no studies have
reported this phenomenon in humans. **Thus, our study is the first to document**
**cholangiocyte contributions to hepatocyte regeneration after PHx.**

Major points

1. The manuscript describes cholangiocyte-to-hepatocyte transdifferentiation as the primary
mechanism of liver regeneration during rapid growth stages. However, only a partial number of
newly regenerated hepatocytes come from her2+ cells at 8 dpi after PHx in Fig.4F and G.

According to the data, the H2B-mGL+ hepatocytes should be the primary cell sources for liver
regeneration (Fig. 4D and F).

The reviewer raises a valid point regarding the representation of our data. In the original
manuscript, the her2 lineage tracing figures were shown as a single confocal plane,
while the CellCousin experiment was represented with a maximum intensity projection
image, which may have led to some confusion. We have now revised the CellCousin
data to display a single confocal plane for consistency and clarity (Fig. 4C), which

clearly shows the restriction of *de novo* hepatocytes (mTagBFP+ cells) to a small region
near the edge of the regenerate.

Additionally, we have quantified the labeling frequency for the *her2* line, finding that it
labels 80.4% of cholangiocytes (Supplementary Fig. 7B). This means we are likely
underestimating the extent of transdifferentiation based on *her2*-lineage tracing alone.
However, by using a knock-in CreER line, we minimize potential non-specific expression
that may arise from a promoter-driven Cre line. Therefore, despite the incomplete
labeling, the *her2* lineage tracing provides the strongest experimental evidence currently
available for cholangiocyte-to-hepatocyte conversion. We believe that Cre/Lox-based
lineage tracing is essential for confirming transdifferentiation, and this approach best
supports our findings.

Furthermore, we would like to emphasize our findings from the cell-cycle analysis
following partial hepatectomy (PHx). Using whole-mount EdU imaging, we found that
EdU+ cells are confined to within 50 μm of the regenerate's edge (Fig. 3D-G; see also
Supplementary Movie 1). This indicates the formation of a localized high-proliferation
zone post-injury, suggesting a spatially focused regenerative response.

Within this proliferative zone, hepatocytes exhibit a significant increase in cell-cycle
activity at 1 dpi, while cholangiocytes show a strong rise in cell-cycle activity at both 1
and 2 dpi. **Remarkably, 20 % of cholangiocytes within 50 μm of the regenerate edge**

**enter S-phase at 1 dpi, highlighting a robust localized response.** Beyond this zone,
neither hepatocytes nor cholangiocytes display any increase in cell-cycle activity,
confirming that the regenerative response is concentrated at the injury site.
**Additionally, we observed that cholangiocytes maintain this proliferative activity**
**longer and more intensely than hepatocytes, indicating a regionally focused role for**
**cholangiocytes in liver regeneration.** These findings underscore the capacity of
cholangiocytes to respond to PHx.

2. Fig.1E and G show that nearly all the newly regenerated hepatocytes (mTagBFP2+) were
derived from non-hepatocytes. Meanwhile, less than half of hepatocytes marked by mCherry-
NTR after 4-OHT treatment were ablated with Mtz incubation (Fig. 1F). What about the other
half of hepatocytes during liver regeneration? Did the mGL+ hepatocyte proliferate, migrate or
die during liver regeneration?

**The reviewer raises an important question regarding whether the spared hepatocytes**
**proliferate in response to partial ablation.** Using whole-mount EdU imaging, we
**quantified cell proliferation across different stages post-injury (Fig. 2B). Our findings**
**reveal that spared hepatocyte proliferation does not significantly increase after partial**
**ablation (Fig. 2C).** Instead, we observed a robust increase in cholangiocyte proliferation
from 6 to 48 hours post-injury (Fig. 2C). Similarly, *de novo* hepatocytes, which are
derived from cholangiocytes, display robust proliferation at 24- and 48-hours post-
partial ablation. These results further support our interpretation that cholangiocytes,
rather than spared hepatocytes, are the primary source of new hepatocytes in our
regeneration model. We have added this data to the revised manuscript and updated

the Results sections to clarify this distinction.

Further, the spared mGL+ hepatocytes persist after partial ablation, as evidenced by
their presence at 5 days post-partial ablation (dppa) in imaging (Fig. 1F) and in single-
cell RNA sequencing data at 9 dppa (Fig. 5D-F).

3. Most de novo regenerated hepatocytes are nls-mTagBFP2+ in Fig. 2F (non-hepatocyte-
derived), but only partial newly regenerated hepatocytes are derived from her2+ cholangiocytes
(Fig.4D-G). These data indicate other non-hepatocyte cell types besides cholangiocytes are also
involved in hepatocyte regeneration. The authors need to address this important point.

We thank the reviewer for raising this important point regarding the potential
involvement of other non-hepatocyte cell types in hepatocyte regeneration. To address
this, we performed an **unbiased pseudotemporal analysis using TSCAN** (1) to
investigate the cellular trajectories during liver regeneration (Figure R1).

The pseudotemporal trajectory of cells during liver regeneration shows a
continuous and unidirectional progression from cholangiocytes (blue) to hepatocytes
(yellow). Importantly, no other cell types were found to exhibit a trajectory toward
hepatocytes, as indicated by the absence of such transitions in the analysis. Cells in
grey, including other non-hepatocyte and non-cholangiocyte populations, do not align
with the pseudotemporal trajectory, supporting the conclusion that these cell types are
not directly involved in the regeneration of hepatocytes. The *in silico* analysis
demonstrates that non-cholangiocytes do not contribute significantly to hepatocyte
regeneration.

Figure R1: Unbiased pseudotemporal analysis identifies cholangiocytes as the source of hepatocytes. (Left) UMAP plot showing the pseudotemporal trajectory during liver regeneration in zebrafish. Cells are colored based on pseudotime, progressing from cholangiocytes (blue) to hepatocytes (yellow), highlighting a continuous transition from cholangiocytes to hepatocytes. No trajectory was identified for the cells in grey. (Right) UMAP plot with clusters annotated by cell type (same as Fig. 5B).

4. The manuscript showed that the inhibition of Notch signalling led to the spontaneous
 emergence of hepatocytes. At the same time, the activities of mTORC1 in the larval
 cholangiocytes was stronger than those in adults. What is the relationship between Notch and
 mTORC1 in regulating cholangiocyte plasticity?

This is an intriguing question. We propose that mTORC1 signaling does not influence
Notch levels under homeostatic conditions. This is evidenced by the fact that in a
growing liver, mTORC1 activity is high (Fig. 7D-F), yet no spontaneous
transdifferentiation occurs. However, upon injury, we hypothesize that elevated
mTORC1 levels enable cholangiocytes to effectively downregulate Notch signaling,
promoting transdifferentiation. This distinction is especially clear when comparing 5 dpf
and 10 dpf larval zebrafish. Previous studies, including our own, indicate that at 5 dpf,
zebrafish are entering a fasting state (2–4), during which mTORC1 activity in
cholangiocytes is low (Supplementary Fig. 10), and PHx does not induce cholangiocyte-
to-hepatocyte transdifferentiation (5).

Based on this, below is a summary of the relationship between mTORC1 levels and
transdifferentiation potential:

Stage	mTORC1 Levels	Cholangiocyte Transdifferentiation upon PHx
5 dpf	Low	Absent [From Zhang et al., iScience , 2021 (5)]
10 dpf	High	Present [This study]
Adult	Low	Absent [From Oderberg and Goessling, JCI insight , 2023 (6)]

Thus, it appears that the mTORC1 pathway primes cholangiocytes to alter their identity
in response to injury. The precise molecular mechanisms underlying this process are
currently unknown and may involve interactions between mTORC1 and injury-induced

signals. We are actively exploring this aspect as the basis for future studies.

Minor points

1. It is essential to assess the ratios of non-hepatocyte-derived hepatocytes among all hepatocytes
during liver regeneration.

For PHx, we have now done this using FACS (Fig. 4E). This analysis showed that at 4

dpi, the *de novo* hepatocytes represent 18.0 ± 2.9 % of hepatocytes in PHx animals, as

opposed to 1.2 ± 0.1 % of hepatocytes in control animals. Note that in PHx, we remove

17.3 % of hepatocytes (Fig. 3B).

For partial ablation, the ratios are presented in Fig. 1F. *De novo* hepatocytes represent

43.5 % of hepatocytes after ablation.

Figure 1F: Percentage of labeled cells after partial ablation. Mean \pm SD of the percentage of hepatocyte labeling before (pre-Mtz, n=14) and after Mtz treatment (0,1 and 5 dppa, n=10 each).

2. Western blot is necessary to demonstrate the expression levels of pS6 in larval and adult
 zebrafish liver, as shown in Fig. 5D.

We have performed western blot and see a stronger signal for pS6 in larval liver as
 compared to the adult liver.

Figure 7F: Western blot analysis of protein expression from whole liver of 10 dpf larvae and female and male adults. The top panel shows the protein levels of pS6 and the bottom panel represents the loading control (GAPDH).

3. Could the inhibition of Notch signalling promote the conversion of cholangiocytes into
 hepatocytes in adult liver after PHx?

By using the exact same pharmacological protocol as utilized for larval zebrafish, the

adult zebrafish did not show any sign of transdifferentiation (Figure R2). We were
 unable to find a single example of overlap between cholangiocyte lineage and
 hepatocytes in adult zebrafish.

Samples were analyzed 2 days post-treatment (dpt). (B) Confocal images of adult liver treated with DMSO and 10 μ M LY411575 for 3 days, showing no overlap or mutual expression between H2B-mGL and H2B-mCherry (n=6/6).

**Reviewer #2 (Remarks to the Author):**

The work by Eski investigates the regenerative response of the liver during a phase of rapid
growth, as seen in young animals. Using zebrafish as a model, they employ two different
ablation strategies, ablation of about 50% hepatocytes throughout the liver and PHx by surgical
lobe resection. Based on genetic lineage tracing and slow histone turn-over tracing, they propose
that new hepatocytes originate from cholangiocytes, rather than remaining hepatocytes or a
mixed origin, as seen in adults or early larval zebrafish. Single cell transcriptomics are employed
to corroborate cholangiocytes trans differentiation as source. Finally, these data further implicate
enhanced mTORC1 signaling as mediator of this cholangiocyte plasticity during heightened
growth phases, supported by subsequent inhibitor treatments.

Overall, the manuscript is well written and supported by imaging of excellent quality and
appropriate quantifications. They also developed interesting new methodologies, including
transgenic animals. The findings are overall novel and exciting, since little is known about
mechanisms of postnatal/-embryonic regeneration. Therefore adding valuable insights to the
hepatology field, and generally very interesting to the regeneration

We thank Reviewer #2 for their positive appraisal of our study and for recognizing the
novelty and relevance of our findings within the field of liver regeneration. We
particularly acknowledge the constructive suggestions related to proliferative response
to injuries, which prompted us to enhance our analysis of proliferative response,
clarifying the contributions of cholangiocytes and hepatocytes to liver regeneration. We
have also revised our imaging data presentation to clearly indicate resection sites.

1. De novo hepatocyte formation after partial hepatocyte ablation exclusively of cholangiocyte
origin is an exciting proposal and contrasts most liver regeneration in mammalian and zebrafish
models, in which healthy hepatocytes contribute a substantial fraction of new cells by
proliferation. To substantiate this intriguing interpretation and this key point of this work,
complementary proliferation studies focusing on the different stages and cell types need to be
performed, because current lineage tracing does for instance not reveal hepatocyte proliferation.

We thank the reviewer for highlighting this aspect of our study and for their suggestion
regarding complementary proliferation studies. In response, we conducted a
quantitative analysis of cell-cycle, using EdU incorporation, to assess cell proliferation
across different stages post-injury (Fig. 2B). **Our findings reveal that spared hepatocyte**
**proliferation does not significantly increase after partial ablation (Fig. 2C).** Instead, we
observed a robust increase in cholangiocyte proliferation from 6 to 48 hours post-injury
(Fig. 2C). Similarly, de novo hepatocytes, which are derived from cholangiocytes, display
robust proliferation at 24- and 48-hours post-partial ablation. These results further
support our interpretation that cholangiocytes, rather than spared hepatocytes, are the
primary source of new hepatocytes in our regeneration model. We have added this data
to the revised manuscript and updated the Results sections to clarify this distinction.

2. Here, it is notable that recombination is already initiated 5-6days before ablation, so that
relatively large clones of NTR-mcherry form (see eg Fig 2A).

What is the recombination pattern and how does regeneration look like when recombination is

induced shortly before NTR-ablation or PHx?

Following 4-OHT treatment, we observe that the default blue fluorescence (nls-
mTagBFP2) persists for approximately 3 days before reaching undetectable levels due
to perdurance of the fluorescent protein, while mCherry-NTR fluorescence takes around
4 days to become robustly detectable (Supplementary Fig. 2C). To ensure clear lineage
distinction, we wait 4.5 days from the end of 4-OHT treatment (5.5 days post-
fertilization) to the induction of injury (10 days post-fertilization). This timing allows for
sufficient transition between fluorescent markers and ensures that the labeled cells are
effectively visualized at the onset of regeneration. We have added this clarification in
the revised manuscript, which would be of interest to researchers who want to adapt
CellCousin model to their organ system.

3. After PHx, which parts of the liver produce new hepatocytes? Surprisingly, this is not clearly
described or shown. This is generally an important point, and specifically because after PHx in
adult zebrafish new cells come either from a blastema or from compensatory growth and the
reasons are not entirely clear yet (Oderberg and Goesling, 2021).

Schematic (Fig2D) indicates that the surgically removed part is replaced by epimorphic
regeneration, in the sense that the missing tissue regrowth is exclusively formed by TagBFP2
cells. is this what is observed? The corresponding experimental results (Fig2F) are difficult to
interpret, since it is unclear where in the image the resection site is located and which part of the
liver is exactly shown.

It should be clearly addressed where in the liver, related to the resection site new hepatocytes

from cholangiocyte transdifferentiation form. Analyzing and quantifying different regions in the
same liver, e.g. left, right lobe and connecting part. Does the right lobe show a compensatory
response contributing new hepatocytes to the regeneration of the liver? Like for the chemical
NTR-model above, complementary proliferation studies are necessary to pinpoint the
contributing of regional sources and cell types. Would that be expected given the 'primed' status
of the cholangiocytes?

This was an important point that we have now addressed. Using whole-mount EdU
imaging, we observed that EdU+ cells are restricted to within **50 µm of the edge of the**
**regenerate** (Fig. 3D-G; see also Supplementary Movie 1). This demonstrates that a
localized region of high proliferation is established after injury, supporting a spatially
focused response.

Within this proliferative zone, hepatocytes show a significant increase in cell-cycle
activity at 1 dpi, whereas cholangiocytes display a robust increase in cell-cycle activity
at both 1 and 2 dpi. **Notably, 20% of cholangiocytes within 50 µm of the regenerate**
**edge are in S-phase at 1 dpi, indicating a strong localized response. Outside of this**
**zone, neither hepatocytes nor cholangiocytes show any increase in cell-cycle activity,**
**underscoring that the regenerative response is concentrated at the injury site.**

Additionally, we find that cholangiocytes sustain this proliferative response longer and
more robustly than hepatocytes, indicating a regionally restricted contribution of
cholangiocytes in liver regeneration. We refer to this area as the "wound border zone" in
the Discussion section.

We have added these findings to the revised manuscript to clarify the spatial dynamics
of hepatocyte and cholangiocyte contributions to regeneration post-PHx.

(In considering terminology, the corresponding author (SPS) was intrigued by the
potential of labeling this region as a "blastema", due to previous experience in zebrafish
fin regeneration. However, while the limb or fin blastema consists of multipotent cells,
this may not apply to liver regeneration. Therefore, for accuracy, we have opted to
describe this area as the "wound border zone," drawing on analogous terminology from
heart regeneration studies (7).)

4. Further to the source of new cells following PHx, where is the resection side in Figure 4D and
F? Where are the groups of mCherry cells located in the provided images in relation to it? Also
please explain why there seem to be small focal clones of hepatocytes derived from
cholangiocytes, rather than a more wider response throughout the remaining organ as one might
expect when larval cholangiocytes show a generally higher plasticity? what is the spatial
distribution of such clones?

We have now outlined the edge of the regenerate in the provided images for clarity. Our
data indicate that cholangiocyte-derived hepatocytes are consistently found in close
proximity to the regenerate edge, in line with our cell-cycle quantification. Regarding the
reviewer's observation on clonality, we also noticed this focal clustering in all of our
images, such as in Figures 4C and 7H. This pattern suggests that a small number of
cholangiocytes may expand to drive regeneration.

Exploring the clonal dynamics of cholangiocytes post-PHx with a multi-color labeling
strategy could provide valuable insights into whether a subset of cholangiocytes
predominantly contribute to regeneration. However, clonal analysis of cholangiocyte
contribution is beyond the scope of our current tools. We have noted this observation in
the revised manuscript and hope to pursue it in future studies.

5. PHx, the impaired growth response of the third lobe is interesting, can the authors exclude that
they do not remove some of the ventral lobe progenitors? Similar right lobe PHx should be
performed as appropriate control for this.

We appreciate the reviewer's suggestion to perform partial hepatectomy (PHx) of the
right lobe as a control for the impaired growth response of the ventral lobe. We
attempted this experiment; however, it resulted in 80% lethality, requiring us to abort the
experiment. We believe the high lethality may stem from the close proximity of the right
lobe to the pancreas, which stores digestive enzymes, as well as to the gall bladder (Fig.
R3). Injury to these adjacent structures likely impacts the animals' survival. In contrast,
the left lobe is more isolated from other hepatopancreatic structures, allowing us to
perform PHx safely in this region.

Figure R3: Proximity of the liver's right lobe to the pancreas and gall bladder. (A) Confocal image showing the proximity between the right liver lobe (red) and pancreatic acinar cells (green). (B) Bright-field image illustrating the right liver lobe positioned above the gall bladder at 10 dpf, highlighting their proximity.

6. Also, please elaborate on the proposed morphological response? is it possible that the anlage for
 the ventral lobe gets damaged during the left lobe surgery – representing the reason for smaller
 lobe?

Length measurement, even if only an approximation of regeneration, suggests that the left lobe
 recovers by an epimorphic mechanisms.

We thank the reviewer for this thought-provoking question. Currently, due to the lack of
 a specific marker for the ventral lobe anlage in zebrafish, we are unable to delineate the
 exact mechanism underlying the observed morphological response. It is indeed
 possible that the anlage of the ventral lobe is directly impacted during left lobe PHx,
 which could explain the smaller size of the ventral lobe during regeneration. We have

added this consideration to the Results section of the revised manuscript (in bold):
“Intriguingly, the ventral lobe remained shorter in the adults that underwent PHx as
compared to controls (Supplementary Fig. 4G, H), suggesting that PHx has a long-term
impact on the morphology of the liver in zebrafish. **However, since specific markers to**
**label the ventral lobe anlage are currently unavailable in zebrafish, we are unable to**
**track the precise origins or status of this anlage post-surgery. It is therefore possible**
**that the ventral lobe anlage is directly impacted or even partially damaged during the**
**left lobe PHx, which could account for its diminished growth.”**

7. The authors make a strong point about unlimited food availability during the growth phase, as
well as increased expression of components of mTORC1 signaling. Is this a real link? It should
be tested whether limited feeding has an effect on the expression of mTORC1 signaling and the
regeneration response.

Since in the late embryo/early larvae hepatocytes also contribute to regeneration, what makes
their response different?

Finally, hepatocytes and cholangiocytes both express mTORC1 targets, what is the role in
hepatocytes in this context? Do the inhibitor treatments give an indication?the authors should at
least discuss possible mechanisms for different functions.

To assess the relationship between feeding, mTORC1 signaling, we performed a 0.25x
feeding regimen and observed a trend toward reduced pS6 levels, though this decrease
was not statistically significant (Fig. R4). The rotifer feed has a high protein content
(approximately 50% (8)), which likely sustains mTORC1 activity despite reduced
feeding.

However, at 5 days post-fertilization (dpf), we and others have shown that the animals
are entering a fasting state (2–4). At this stage, the pS6 levels are significantly low (Fig.
R4). At 5 dpf, it has been shown that PHx at this stage does not induce
transdifferentiation (5).

This suggests that mTORC1 activation promotes cholangiocyte plasticity.

The role of mTORC1 in hepatocytes likely revolves around their critical functions in
protein processing and metabolism, given that mTORC1 activation is highly responsive

to protein uptake (9). Particularly, at 10 dpf, the animals are being fed with rotifers, a
high protein diet (44 – 50 % protein content (8)). It would be of interest to raise
zebrafish on diets varying in protein and lipid content and to evaluate mTORC1 levels
and the cellular source of liver regeneration. This would be more informative rather than
reducing the level of feed, as reduction in rotifer feed did not impact pS6 levels (Fig. R4).
In the future, we can attempt to standardize alternative feeding regiment to investigate
this question. We have added this point to the Discussion section of the manuscript.

8. Given the use of the her2-driver line in this study, validation, including quantification, of cell
type specificity should be provided (similar to figS8).

The same should be done for the her9-driver line. The images are excellent, though unclear
which other cell types are marked and at which frequency.

We have quantified the labeling frequency for both the her2 and her9 lines to validate
cell type specificity. For the her2 line, we found that it labels 80.4% of cholangiocytes
(Supplementary Fig. 7A-B). Within the her2-lineage cells, 93.7% were identified as
cholangiocytes (Supplementary Fig. 7C)). Thus, the her2 lineage is predominantly
specific to cholangiocytes in the liver.

For the her9 line, it labels 59.3% of cholangiocytes (Supplementary Fig. 8A-C). Within the
her9 lineage, 60.2% are cholangiocytes (Supplementary Fig. 8D). Additionally, our single-
cell RNA-Seq dataset reveals her9 expression in hepatic stellate cells (Fig. 5C). Using
the *TgBAC(hand2:EGFP)* line, which specifically labels hepatic stellate cells, we
identified that 30.0% of her9-lineage cells are stellate cells (Supplementary Fig. 8D, E).

It is important to note that our her2-lineage tracing demonstrates that cholangiocytes
contribute to hepatocyte regeneration. While we do not propose *her9* as a specific
cholangiocyte marker, as noted in the manuscript, it labels multiple cell types in the liver
parenchyma. However, *her9* expression is completely excluding from hepatocytes.
Thus, *her9*-lineage tracing indicates that non-hepatocytes can generate hepatocytes.
We have added these quantifications to the revised manuscript to further clarify the cell
type specificity and functionality of each driver line.

9. Text and Figure S8: states that the her9+ lineage mainly contains cholangiocytes. Such a
statement should be supported by appropriate quantification, which is essential for the
conclusions that use this line.

Please see the response to the previous question. In the text, we have removed the
word “mainly” and added the appropriate quantifications.

Minor points:

1. Fig4D,F – please show consistently the larger magnifications.

We have edited the figure to have larger magnifications.

2. For the cholangiocyte response, the authors use several times the term reprogramming,
which is generally associated with the removal or remodeling of epigenetic marks, which they do
not address. Hence consistent use of transdifferentiation seems more appropriate.

We have removed reprogramming from the manuscript. And used transdifferentiation

throughout the text.

3. Please indicate what the data in Figure 1G, 2G and S7 E,G are normalized to.

All data is normalized to volume. We mention this in the Methods Section and have also
included this in the figures now.

4. Figure 4: mention of non-existing panel H in Figure legends.

Apologies for the error. We have corrected the figure legend.

5. Fig4D,F – indicate the resection site.

We added the edge of the regenerate with dotted line. It is difficult to identify the
resection site as there aren't morphological landmarks to delineate the injury site. Thus,
we can only identify the edge of the regenerating liver.

6. For a number of imaging data, it would be very helpful to see magnifications of the data, e.g.
fig 2F or 5G,

Thank you for the suggestion. We have edited the figure panels in two ways:

- 1. Included high-magnification images
2. Edited the images from z-stack to a single plane for confocal images to increase
clarity.

These edits dramatically improved the image presentation. We acknowledge the
reviewer's feedback to improve the presentation.

References:

1. Ji Z, Ji H. TSCAN: Pseudo-time reconstruction and evaluation in single-cell RNA-seq
analysis. *Nucleic Acids Res.* 2016;44:e117.

2. Macarena Pozo-Morales, Ansa E Cobham, Cielo Centola, Mary Cathleen McKinney,
Peiduo Liu, Camille Perazzolo, et al. Starvation resistant cavefish reveal conserved
mechanisms of starvation-induced hepatic lipotoxicity. *bioRxiv.* 2024;2024.01.10.574986.

3. Pozo-Morales M, Garteizgogea I, Perazzolo C, So J, Shin D, Singh SP. In vivo imaging
of calcium dynamics in zebrafish hepatocytes. *Hepatology.* 2023;77:789.

4. Xu H, Jiang Y, Miao X-M, Tao Y-X, Xie L, Li Y. A Model Construction of Starvation
Induces Hepatic Steatosis and Transcriptome Analysis in Zebrafish Larvae. *Biology.*
2021;10:92.

5. Zhang W, Chen J, Ni R, Yang Q, Luo L, He J. Contributions of biliary epithelial cells to
hepatocyte homeostasis and regeneration in zebrafish. *iScience.* 2021;24:102142.

6. Oderberg IM, Goessling W. Biliary epithelial cells are facultative liver stem cells during
liver regeneration in adult zebrafish. *JCI Insight [Internet].* 2023 [cited 2023 Jun 18];8.
Available from: <https://insight.jci.org/articles/view/163929>

7. Constanty F, Wu B, Wei K-H, Lin I-T, Dallmann J, Guenther S, et al. Border-zone
cardiomyocytes and macrophages contribute to remodeling of the extracellular matrix to
promote cardiomyocyte invasion during zebrafish cardiac regeneration. *BioRxiv Prepr.*
*Serv. Biol.* 2024;2024.03.12.584570.

8. Srivastava A, Hamre K, Stoss J, Chakrabarti R, Tonheim SK. Protein content and amino
acid composition of the live feed rotifer (*Brachionus plicatilis*): With emphasis on the water
soluble fraction. *Aquaculture*. 2006;254:534–543.

9. Jewell JL, Russell RC, Guan K-L. Amino acid signalling upstream of mTOR. *Nat. Rev.*
*Mol. Cell Biol.* 2013;14:133–139.

*Cholangiocytes contribute to hepatocyte regeneration after partial liver*
*injury during growth spurt in zebrafish.*

**Sema Elif Eski, Jiarui Mi, Macarena Pozo-Morales, Gabriel Garnik Hovhannisyan,**
**Camille Perazzolo, Rita Manco, Imane Ez-Zammoury, Dev Barbhaya, Anne Lefort,**
**Frédéric Libert, Federico Marini, Esteban N. Gurzov, Olov Andersson, Sumeet Pal**
**Singh***

* Corresponding author: sumeet.pal.singh@ulb.be

We appreciate the constructive feedback provided by the reviewers. We have carefully
considered the comments and have made substantial revisions to strengthen the
manuscript. Below, we address each concern in detail, incorporating new data,
clarifications, and modifications to improve the rigor and clarity of our study.

For the revision, our response is in blue. Figures specific to the rebuttal letter are labeled
as Figure Rn.

**REVIEWER COMMENTS**

**Reviewer #1 (Remarks to the Author):**

Again, I would repeat the key point of my original comments: The cholangiocyte-to-hepatocyte
transdifferentiation has been intensively studied so that this manuscript is devoid of novelty.

Besides, the fact that this transdifferentiation only occurs in larvae, but not in adult zebrafish and
mice after PHx, diminishes the significance of this finding.

We acknowledge the reviewer's concerns regarding the novelty of our findings. While it
is true that cholangiocyte-to-hepatocyte transdifferentiation has been studied in adult
models, our work provides new insights into the context of liver regeneration during
growth spurts. Unlike previously characterized adult systems, we demonstrate that this
process occurs robustly in growing zebrafish larvae even when a significant number of
hepatocytes remain after injury, challenging the prevailing model that
transdifferentiation occurs only under conditions of severe hepatocyte loss. This
highlights a previously underappreciated role of cholangiocyte plasticity in regeneration.

Furthermore, while transdifferentiation is absent in adult zebrafish and adult
mice after partial hepatectomy, its presence in growing zebrafish larvae suggests that
**developmental stage plays a key role in dictating cellular plasticity** .

Mouse Data (Redacted in Final Version)

Regarding the lack of evidence for this phenomenon in mice (CD1 background), we have
conducted preliminary analyses in juvenile mice after PHx. First, we established a PHx
model for juvenile mice (Fig. R1). Next, we performed partial hepatectomy, removing
50% of the liver in postnatal day 21 (P21) mice. We chose P21 mice because they are
still in early postnatal development (1), not yet weaned, and their livers are actively
maturing, with a higher percentage of proliferating hepatocytes compared to adult mice
(2,3).

To assess potential cholangiocyte-to-hepatocyte conversion, we harvested the
liver at postoperative day 2 and performed double immunohistochemistry for CK19 and
HNF4 α . We specifically looked for hepatocytes that co-expressed CK19 and HNF4 α ,
which would indicate a transitional state between cholangiocytes and hepatocytes. Our
analysis revealed the presence of CK19+/HNF4 α + hepatocytes within regenerating liver
(Fig. R2A), indicating that a subset of hepatocytes may arise from cholangiocyte in
juvenile mice following PHx. Notably, these cells appear despite the fact that native
hepatocytes remain highly proliferative after PHx (Fig. R2B).

Figure R1. Schematic and representative images of the partial hepatectomy (PHx) procedure in juvenile mice.

(Left) Equipment setup for PHx, including an anesthesia machine (top) and a surgical microscope (bottom).

(Center) Stepwise surgical procedure:

- (a) Induction of anesthesia and positioning of the juvenile mouse.
- (b) Laparoscopy to access the abdominal cavity.
- (c) Exposure of the left lateral liver lobe, which is targeted for resection.
- (d) Surgical resection of the left lateral lobe.
- (e) Isolated resected liver lobe post-PHx.

Schematic illustration of liver anatomy, indicating the location of the left lateral lobe, which is surgically removed during the procedure.

(Right) Liver morphology following PHx. P21 liver prior to surgery (top). Liver remnant on post-operative day 2 (POD2), illustrating the remaining liver tissue after partial resection (bottom).

These findings, similar to our observations in zebrafish, **challenge the notion**

**that cholangiocytes-to-hepatocytes transdifferentiation is strictly limited to adult**

**individuals and occurs only when hepatocytes proliferation is impaired** . Instead, our

data suggest that under specific conditions, such as postnatal liver development,

cholangiocytes may contribute to hepatocyte regeneration even when hepatocyte-driven

repair is still active. These findings support our hypothesis that **hepatic plasticity is**

**enhanced during early postnatal growth** and that cholangiocytes may play a more

active role in liver regeneration when hepatocyte proliferation alone is insufficient. We
acknowledge that definitive proof of transdifferentiation in mice requires lineage-tracing
approaches, we are currently validating this observation using cholangiocyte-specific
lineage tracing (osteopontin-iCreERT2 (4,5)) and single-nucleus RNA sequencing
(snRNA-seq). These experiments aim to determine whether cholangiocytes directly
contribute to hepatocyte regeneration in the postnatal liver. However, this investigation
requires a comprehensive, standalone study and is beyond the scope of the current
manuscript.

(End of Redaction)

Furthermore, some key tools used in this study are not optimal. For example, the her9 and her2
transgenes, because they label not only cholangiocytes but also other cell types such as satellite
cells.

Regarding the specificity of our lineage-tracing tools, we would like to clarify that the
*her2* transgene is specific to cholangiocytes (Fig. 5C) and does not label other liver cell
types. Our validation experiments (Supplementary Fig. 7) show that **her2+ lineage cells**
**predominantly mark cholangiocytes (93.7%)** , with minimal labeling of other cell types.
This specificity strengthens our conclusions regarding the cholangiocyte origin of
hepatocytes during regeneration.

Therefore, I would not support publication of this paper at Nature Communications.

**Reviewer #3 (Remarks to the Author):**

The authors addressed several prior reviewers' comments and made additional important
observations. These new data strengthen the paper and enhance its novelty. However, I have
additional questions and comments:

1. Cholangiocyte Contribution to Hepatocyte Regeneration

The main conclusion of the paper is that cholangiocytes contribute to hepatocyte regeneration
during growth spurts in zebrafish, even when a significant number of hepatocytes remain after
injury. Cholangiocytes are heterogeneous based on their localization, morphology, and gene
signatures. Which cholangiocyte population(s) specifically contribute to hepatocyte regeneration
in the two injury models described in the paper? Furthermore, the current data do not provide
sufficient evidence to exclude the possibility that hepatic progenitor cells, rather than
differentiated cholangiocytes, contribute to this regeneration. Additional experiments should be
conducted. Otherwise, the main conclusion from this paper is too strong of a statement.

*[The following Additional comment by the Reviewer was moved here by authors.]*

Additional comments regarding critique 1 (Cholangiocyte Contribution to Hepatocyte
Regeneration):

To date, all published work on zebrafish liver regeneration involving cholangiocyte-to-
hepatocyte transdifferentiation relies on Tp1-Cre or Tp1-CreERT2 to lineage trace existing
cholangiocytes. However, Tp1, a Notch-responsive element, is not a cholangiocyte-specific

promoter. It can also label hepatic progenitor cells. Additionally, the cholangiocyte markers
commonly used in zebrafish studies, such as 2F11, *alcam*, and *epcam*, are not exclusively
expressed in differentiated cholangiocytes. This represents a significant limitation of these
studies, which should be properly addressed.

Determining whether hepatic progenitors or differentiated cholangiocytes give rise to
regenerating hepatocytes would require the development of additional tools and may be beyond
the scope of this paper. However, I believe the single-cell RNA-seq data presented by the authors
could provide valuable insights into this question. Within the identified “cholangiocyte”
population, are there distinct subgroups? Specifically, are there more differentiated
cholangiocyte subpopulations that express cholangiocyte-specific transporters and metabolic
genes? Conversely, are there “cholangiocytes” that appear less mature and exhibit more
progenitor-like properties?

Acknowledging the heterogeneity within *Tp1+* populations and recognizing that different subsets
may contribute differently to hepatocyte regeneration is a critical first step toward understanding
the key cellular players in this process. Thus adding additional analysis of the scRNA-seq data
will strengthen the paper.

We appreciate the reviewer’s insightful comments regarding the heterogeneity of the
*Tp1+* cholangiocyte population and the importance of further dissecting the cellular
origins of transdifferentiation. Recent work from Jiarui Mi, Olov Andersson et al.,
currently posted on *bioRxiv* (6), has provided a refined classification of cholangiocyte

subpopulations in the zebrafish liver, identifying three distinct stages of maturity:
Intrahepatic Duct (IHD) → Intermediate Duct → Luminal Duct. In this framework, IHD
and intermediate duct cells are labeled by *her2*, while luminal ductal cells express *krt4*,
representing the most mature cholangiocytes.

To further investigate this heterogeneity, we performed sub-clustering analysis
on our control scRNA-seq dataset and identified four distinct cholangiocyte clusters:
one enriched for *krt4* (luminal ductal cells), two clusters containing a mix of IHD and
intermediate duct cells, and a proliferative cholangiocyte cluster (Supplementary Fig.
6A-E). To delineate the trajectory of cholangiocyte transdifferentiation, we conducted
pseudotemporal analysis using Monocle, which revealed that IHD and intermediate duct
cells are the primary source of transdifferentiation (Supplementary Fig. 6F-H).

Additionally, the results of the pseudotemporal analysis align well with our
experimental findings using Cre/Lox-based lineage tracing with *her2-CreER*,
demonstrating that IHD and intermediate duct cells contribute to hepatocyte
regeneration upon injury, but not under homeostatic conditions. Importantly, these cells
represent a bona fide cholangiocyte lineage and do not contribute to hepatocytes in the
absence of injury.

In summary, our data suggests that transdifferentiation is driven by less mature
cholangiocytes (IHD and intermediate duct cells). Future studies will be required to
determine whether luminal cholangiocytes possess the capacity to alter their identity
under specific conditions or if their fate is terminally differentiated. Thus, we have
added this to the Discussion:

“Within the cholangiocyte population, our pseudotemporal analysis revealed that
intrahepatic (IHD) and intermediate cholangiocytes serve as the primary source of
cholangiocyte-to-hepatocyte transdifferentiation, while luminal cholangiocytes,
represented by *krt4* expression, do not appear to contribute (Supplementary Fig. 6).
Future studies will be needed to determine whether luminal cholangiocytes retain the
capacity to alter their identity under specific conditions or if they are terminally
differentiated.”

2. Comparative Analysis of Partial Hepatectomy (PHx)

The authors show that partial hepatectomy (PHx) induces cholangiocyte transdifferentiation at
10 days post-fertilization (dpf), whereas previous studies reported that PHx at 5 dpf or in adult
zebrafish does not induce cholangiocyte transdifferentiation. It is important to point out that
different transgenic zebrafish models were used in these studies, which may explain the differing
outcomes.

We appreciate this point and have now explicitly addressed the differences in
transgenic models used across studies in the Discussion Section:

“At 5 dpf, it has been shown that PHx does not induce cholangiocyte-to-hepatocyte
transdifferentiation, similar to adult zebrafish. Notably, previous studies utilized
*Tg(tp1:CreER)* for lineage tracing of cholangiocytes in 5 dpf larval and adult zebrafish,
which may have influenced the observed absence of transdifferentiation.”

3. Control for PHx Experiments

What control was used in the PHx experiments? A sham control should be included to ensure
that the surgical incision and removal of the skin do not influence ventral lobe growth
independently of the hepatectomy.

A sham control was performed to account for potential effects of the incision or skin
removal. The data, presented in Supplementary Fig. 4I, demonstrate that the incision
alone does not influence ventral lobe growth.

4. Measurement of Liver Size

The exclusion of compensatory growth after PHx is primarily based on lobe length
measurements (Figure 4D–H). However, liver area or volume measurements would be more
appropriate to assess liver size accurately.

We agree that liver volume measurements provide a more comprehensive assessment.

While our imaging setup primarily allowed for 2D measurements, we have now
performed additional volumetric quantifications at 11 dpi and updated the results in
Supplementary Fig. 4J. The new quantifications demonstrate a significant reduction in
liver volume after PHx as compared to uninjured or sham injured controls, further
supporting our conclusions.

5. Identification of Her9-Lineage Cells

The authors concluded that 30% of her9-lineage cells are hepatic stellate cells. However, in
Figure S8E, it is evident that all hand2:GFP+/Her9+ double-positive cells are located at the liver

surface within the mesothelium. These cells are not hepatic stellate cells. The authors should
revise this conclusion accordingly.

We agree with the reviewer. *hand2+* cells include hepatic stellate cells, mesothelial
cells, and perivascular cells. To ensure accuracy, we have revised the text and figure
legend to describe the overlap between *her9*-lineage and *hand2+* cells without
assigning a specific identity to these cells.

“Additionally, our single-cell RNA-Seq dataset revealed that *her9* is expressed in *hand2+*
cells (Fig. 5C). Using the *TgBAC(hand2:EGFP)* line, which labels hepatic stellate cells,
mesothelial cells and perivascular cells (7), we identified that 30.0% of *her9*-lineage
cells were *hand2+* (Supplementary Fig. 9D, E).”

**References:**

1. Brust V, Schindler PM, Lewejohann L. Lifetime development of behavioural phenotype in
the house mouse (*Mus musculus*). *Front. Zool.* 2015;12:S17.

2. Liang Y, Kaneko K, Xin B, Lee J, Sun X, Zhang K, et al. Temporal analyses of postnatal
liver development and maturation by single-cell transcriptomics. *Dev. Cell.* 2022;57:398-
414.e5.

3. Septer S, Edwards G, Gunewardena S, Wolfe A, Li H, Daniel J, et al. Yes-associated
protein is involved in proliferation and differentiation during postnatal liver development.
*Am. J. Physiol. - Gastrointest. Liver Physiol.* 2012;302:G493–G503.

4. Lesaffer B, Verboven E, Van Huffel L, Moya IM, van Grunsven LA, Leclercq IA, et al.
Comparison of the Opn-CreER and Ck19-CreER Drivers in Bile Ducts of Normal and
Injured Mouse Livers. *Cells.* 2019;8:380.

5. Manco R, Clerbaux L-A, Verhulst S, Bou Nader M, Sempoux C, Ambroise J, et al.
Reactive cholangiocytes differentiate into proliferative hepatocytes with efficient DNA
repair in mice with chronic liver injury. *J. Hepatol.* 2019;70:1180–1191.

6. Mi J, Ren L, Liu K-C, Buttò L, Colquhoun D, Andersson O. Deciphering the zebrafish
hepatic duct heterogeneity and cell plasticity using lineage tracing and single-cell
transcriptomics [Internet]. 2025 [cited 2025 Feb 19];2025.01.09.631719. Available from:
<https://www.biorxiv.org/content/10.1101/2025.01.09.631719v1>

7. Yin C, Evason KJ, Maher JJ, Stainier DY. The bHLH Transcription Factor Hand2 Marks
Hepatic Stellate Cells in Zebrafish: Analysis of Stellate Cell Entry into the Developing
Liver. *Hepatology*. Baltim. Md. 2012;56:1958.

*Cholangiocytes contribute to hepatocyte regeneration after partial liver*
*injury during growth spurt in zebrafish.*

**Sema Elif Eski, Jiarui Mi, Macarena Pozo-Morales, Gabriel Garnik Hovhannisyan,**
**Camille Perazzolo, Rita Manco, Imane Ez-Zammoury, Dev Barbhaya, Anne Lefort,**
**Frédéric Libert, Federico Marini, Esteban N. Gurzov, Olov Andersson, Sumeet Pal**
**Singh***

* Corresponding author: sumeet.pal.singh@ulb.be

We thank the reviewer for their continued engagement and thoughtful feedback. We
have carefully considered each comment and have made additional revisions and
clarifications to strengthen the manuscript. Below, we address the concerns in detail,
incorporating new analyses and text modifications aimed at improving the rigor, clarity,
and interpretability of our study.

For the revision, our response is in blue.

**REVIEWER COMMENTS**

**Reviewer #3 (Remarks to the Author):**

The author's responses to reviewers' critiques have raised more questions as detailed below.

1. Newly generated data on liver regeneration after partial hepatectomy in juvenile mice.

The observation in Figure R2 neither supports nor refutes the authors' conclusion that

cholangiocytes contribute to hepatocyte regeneration in juvenile mice following PHx. The

percentage of CK19;HNF4 α double-positive hepatocytes is not reported, nor is the number of

animals analyzed. Moreover, in the absence of lineage tracing, the presence of these double-

positive cells could also suggest that hepatocytes contribute to cholangiocyte regeneration after

PHx.

We thank the reviewer for this important clarification. As noted in our previous revision,

the mouse data were included **solely in response to Reviewer #1**, who had questioned

whether the observed regenerative mechanism might be specific to zebrafish. We wish

to emphasize again that **these mouse experiments are preliminary**, and we agree fully

with the reviewer that lineage tracing is essential to draw any definitive conclusions.

We have only recently established the PHx procedure in postnatal mice and

presented this early analysis to highlight a potential direction for future investigation. As

the reviewer #3 rightly points out, the detection of CK19+/HNF4 α + cells without lineage

tracing does not allow us to distinguish between cholangiocyte-to-hepatocyte or

hepatocyte-to-cholangiocyte conversion. We are currently pursuing a dedicated study

involving Opn:CreER-based lineage tracing and single-nucleus RNA sequencing to

resolve this question, but those experiments are outside the scope of the current
manuscript.

Finally, these preliminary data are not part of the manuscript under
consideration. However, we want to highlight that liver regeneration in the juvenile
mouse remains an underexplored area, and we believe our new setup opens important
avenues for comparative investigation.

2. The authors presented new scRNA-seq data demonstrating four cholangiocyte clusters in the
control zebrafish liver: one enriched for krt4 (representing luminal ductal cells), two clusters
comprising a mix of intrahepatic duct (IHD) and intermediate ductal cells, and a proliferative
cholangiocyte cluster. Using pseudotemporal analysis, they proposed that IHD and intermediate
ductal cells are the primary source of transdifferentiation.

However, the data in Supplemental Figure 6 raises more questions. In Fig. 6F, the
“cholangiocytes rest” cluster (grey) appears to be more poised to transdifferentiate to
hepatocytes than the other four cholangiocyte clusters, yet this cluster is not discussed in the text.

We thank the reviewer for pointing out the “cholangiocytes rest” and apologize for the
lack of clarification. The “cholangiocytes rest” cluster corresponds to cholangiocytes
derived from injured samples. Our subclustering analysis was specifically performed on
control (uninjured) samples to identify distinct cholangiocyte states under uninjured
conditions. These subclusters were then used to trace which populations in the
uninjured liver might give rise to hepatocytes after injury.

However, to perform pseudotemporal trajectory analysis, the dataset must

contain a continuous sequence of states—from the starting population, through
intermediate/transitional states (which arise only after injury), to the end point.
Therefore, pseudotime inference was necessarily conducted using the entire scRNA-seq
dataset, including both uninjured and injured samples.

Within this integrated dataset, the “cholangiocytes rest” cluster emerges from
injured livers and represents a more heterogeneous population, which includes the
transitional state. Our aim in the pseudotemporal analysis was to identify which
uninjured subpopulation was most likely to initiate the transition toward hepatocyte
identity. For this reason, we focused our analysis and interpretation on the uninjured
cholangiocyte subclusters.

We have now updated the Methods and Figure Legend for Supplementary Figure
6 as follows to clarify this point. (**Edits in Bold**)

Methods

Subclustering and pseudotime trajectory analysis

For subclustering, cholangiocytes from uninjured (control) samples were subsetted
from the dataset and processed following standard normalization procedures.
Dimensionality reduction was performed by selecting the top 2,000 highly variable
genes, followed by principal component analysis using the top 22 principal
components. A neighborhood graph was computed with ‘n_neighbors=15’, and
clustering was performed at a resolution of 0.4. Clusters enriched in acinar cell-specific
markers (indicative of contamination) and those exhibiting high ribosomal and heat-
shock gene expression (potentially damaged cells) were excluded from further analysis.

To reconstruct the differentiation trajectory, we used Monocle3 ⁷⁶ **on the**
**combined dataset of injured and uninjured livers, as pseudotemporal inference**
**requires a continuum of cellular states that are not all present in control samples**
**alone. Specifically, transitional or intermediate states appear only upon injury and**
**were essential for trajectory learning.** Clustering was performed using the
*cluster_cells()* function with a resolution parameter of 1e-4, and trajectory graph learning
was conducted using the *learn_graph()* function with 'minimal_branch_len=10'. Cells
were then ordered along the inferred trajectory using the *order_cells()* function with
default parameters. The root node for pseudotime inference was manually defined
within the IHD/intermediate cholangiocyte cluster from control samples to ensure
biologically relevant reconstruction of the transdifferentiation process.

For visualization of subclusters within the cholangiocyte population, UMAP
coordinates computed in Scanpy were used in place of the default Monocle3 layout to
maintain consistency across all analyses. **The “cholangiocytes rest” cluster refers to**
**the population of cholangiocytes derived from injured samples.**

Supplementary Figure 6 Legend

(F) UMAP visualization **of integrated single-cell data from both uninjured and injured**
**livers, including cholangiocyte subclusters from controls (colored) and**

**“cholangiocytes rest” (grey) from injured samples.** Hepatocytes are shown in green.

(G) Pseudotime trajectory inferred using Monocle, showing progression from

cholangiocytes to hepatocytes. **Analysis was performed on the full integrated dataset,**

**including both uninjured and injured samples, to capture transitional states absent**
**from control livers.**

The authors also refer to their previous scRNA-seq study of the adult zebrafish liver, in which
IHD and intermediate ductal cells were also identified. Notably, PHx at the adult stage does not
lead to cholangiocyte-to-hepatocyte transdifferentiation. Comparing the transcriptomes of IHD
and intermediate ductal cells in juvenile versus adult livers would be crucial for uncovering the
distinct mechanisms underlying liver regeneration at these two stages.

We thank the reviewer for this insightful suggestion. In our original analysis, we had
indeed performed a comparative transcriptomic analysis between cholangiocytes from
13 dpf larval and 13 mpf adult uninjured livers (Fig. 7A, B), which led to the identification
of the mTORC1 pathway as being upregulated in larval cholangiocytes (Fig. 7C). This
was validated by immunofluorescence and western blot analysis (Fig. 7D-F)
Functionally, we showed that activation of this pathway in the uninjured context was
sufficient to potentiate cholangiocyte-to-hepatocyte transdifferentiation after injury (Fig.
7G-I), highlighting a key regulator of regenerative plasticity.

However, as the reviewer correctly notes, our prior analysis was conducted on
the entire cholangiocyte population, without distinguishing specific subtypes such as
IHD/intermediate and luminal cholangiocytes.

To directly address this point, we have now performed a focused comparative
analysis between the IHD/intermediate and luminal cholangiocyte clusters isolated
from uninjured larval and adult zebrafish livers (Supplementary Fig. 11). This refined
analysis identified 4415 genes significantly enriched in larval IHD/intermediate

cholangiocytes (Supplementary Table 2), including multiple components of the
mTORC1 signaling pathway (Supplementary Fig. 11D). Genes related to mTORC1
pathway were also upregulated in the luminal cholangiocytes from larval zebrafish
(Supplementary Fig. 11D). These results validate and strengthen our original findings,
now confirming that the differential activation of mTORC1 signaling is already present
at the level of the IHD/intermediate cholangiocytes, which are potentially most relevant
to transdifferentiation.

We have updated the Results section to reflect this new analysis and added the
corresponding data to Supplementary Figure 11. The edits (in bold) are listed below:

Results (Page 17)

Furthermore, larval cholangiocytes exhibited upregulation of genes associated with the
mTORC1 signalling pathway (Fig. 7C), which has been reported to be critical for the
transdifferentiation of cholangiocytes⁵²⁻⁵⁴. **We also performed differential gene**
**expression analysis of cholangiocyte subclusters (Supplementary Fig. 11A-C).**
**Differentially expressed genes between larval and adult stage corresponding to the**
**IHD/intermediate and luminal cholangiocytes are provided in Supplementary Table 2**
**and Supplementary Table 3, respectively. This analysis revealed that mTORC1**
**pathway activity-related genes are elevated in both IHD/intermediate and luminal**
**cholangiocytes from larval livers compared to adults (Supplementary Fig. 11D-E).**

Supplementary Figure 11

A Cholangiocytes - adult

B Cholangiocytes - adult and larvae

C

D

Supplementary Figure 11: Comparative transcriptomic analysis of cholangiocyte subclusters from larval and adult zebrafish livers.

(A) UMAP visualization of cholangiocytes from uninjured 18 mpf adult zebrafish liver, showing subclustering into intrahepatic duct (IHD)/intermediate (blue) and luminal (orange) populations. **(B)** Integrated UMAP plot of cholangiocytes from 13 dpf larval and adult livers, color-coded by subcluster and developmental stage: Adult-IHD/Intermediate (blue), Adult-Luminal (orange), Larval-IHD/Intermediate (green), Larval-Luminal (red), and Larval-Proliferative (purple). **(C)** Violin plots showing expression of representative marker genes across cholangiocyte subclusters from larval and adult livers, including *anxa4* (general cholangiocyte), *her2* (IHD/intermediate), *cdh1* (luminal), and *mki67* (proliferative). **(D)** Top: Heatmap showing average expression of selected genes associated with mTORC1 signaling and cholangiocyte plasticity across each cholangiocyte subcluster. Bottom: Bubble plot showing differentially expressed genes (DEGs) related to the mTORC1 pathway between larval and adult cholangiocytes. Dot size represents the adjusted p-value, and color intensity reflects log₂ fold change.